

# X-BASE: the first terrestrial carbon and water flux products from an extended data-driven scaling framework, FLUXCOM-X

Jacob A Nelson[1,*], Sophia Walther[1,*], Fabian Gans[1], Basil Kraft[1], Ulrich Weber[1], Kimberly Novick[2], Nina Buchmann[3], Mirco Migliavacca[4], Georg Wohlfahrt[5], Ladislav Šigut[6], Andreas Ibrom[7], Dario Papale[8,9], Mathias Göckede[1], Gregory Duveiller[1], Alexander Knohl[10,11], Lukas Hörtnagl[3], Russell L. Scott[12], Weijie Zhang[1], Zayd Mahmoud Hamdi[1], Markus Reichstein[1,13], Sergio Aranda-Barranco[14,15], Jonas Ardö[16], Maarten Op de Beeck[17,18], Dave Billesbach[19], David Bowling[20], Rosvel Bracho[21], Christian Brümmer[22], Gustau Camps-Valls[23], Shiping Chen[24], Jamie Rose Cleverly[25], Ankur Desai[26], Gang Dong[27], Tarek S. El-Madany[1], Eugenie Susanne Euskirchen[28], Iris Feigenwinter[3], Marta Galvagno[29], Giacomo Al. Gerosa[30], Bert Gielen[31], Ignacio Goded[4], Sarah Goslee[32], Christopher Michael Gough[33], Bernard Heinesch[34], Kazuhito Ichii[35], Marcin Antoni Jackowicz-Korczynski[36,37], Anne Klosterhalfen[38], Sara Knox[39,40], Hideki Kobayashi[41], Kukka-Maaria Kohonen[3], Mika Korkiakoski[42], Ivan Mammarella[43], Gharun Mana[44], Riccardo Marzuoli[45], Roser Matamala[46,47,48], Stefan Metzger[49,50], Leonardo Montagnani[51], Giacomo Nicolini[9], Thomas O'Halloran[52,53], Jean-Marc Ourcival[54], Matthias Peichl[55], Elise Pendall[56], Borja Ruiz Reverter[57], Marilyn Roland[58], Simone Sabbatini[9,59], Torsten Sachs[60], Marius Schmidt[61], Christopher R Schwalm[62], Ankit Shekhar[3], Richard Silberstein[63], Maria Lucia Silveira[64], Donatella Spano[65,9], Torbern Tagesson[66,67], Gianluca Tramontana[68], Carlo Trotta[9], Fabio Turco[3], Timo Vesala[69,70], Caroline Vincke[71], Domenico Vitale[72], Enrique R. Vivoni[73,74], Yi Wang[3], William Woodgate[75,76], Enrico A. Yepez[77], Junhui Zhang[78,79], Donatella Zona[80], and Martin Jung[1]

[*]These authors contributed equally to this work.
[1]Max-Planck-Institute for Biogeochemistry, Germany
[2]O'Neill School of Public and Environmental Affairs, Indiana University - Bloomington (USA)
[3]Department of Environmental Systems Science, ETH Zurich, 8092 Zurich, Switzerland
[4]European Commission, Joint Research Centre, Ispra (Varese), Italy
[5]Universität Innsbruck, Institut für Ökologie, Sternwartestr. 15, 6020 Innsbruck, AUSTRIA
[6]Global Change Research Institute CAS, Bělidla 986/4a, CZ-60300 Brno, Czech Republic
[7]Department of Environment and Resource Engineering, Technical University of Denmark (DTU), Bygningstorvet 115, 2800 Kgs. Lyngby, Denmark
[8]Research Institute on Terrestrial Ecosystems (IRET), National Research Council (CNR), 00010 Montelibretti (Roma), Italy
[9]CMCC Foundation - Euro-Mediterranean Center on Climate Change, Italy
[10]University of Goettingen, Bioclimatology, Faculty of Forest Sciences, 37077 Göttingen, Germany.
[11]University of Goettingen, Centre of Biodiversity and Sustainable Land Use (CBL), 37077 Göttingen, Germany
[12]Southwest Watershed Research Center, USDA-ARS, Tucson, AZ
[13]German Centre for Integrative Biodiversity Research (iDiv) Halle–Jena–Leipzig, Leipzig, Germany
[14]Department of Ecology, University of Granada, Granada, Spain
[15]Andalusian Institute for Earth System Research (CEAMA-IISTA), University of Granada, Granada, Spain
[16]Department of Physical Geography and Ecosystem Science Sölvegatan 12 223 62 Lund, Sweden
[17]Research Group Plants and Ecosystems, Department of Biology, University of Antwerp
[18]ICOS / Ecosystem Thematic Centre
[19]University of Nebraska-Lincoln, School of Natural Resources (Retired)



[20]School of Biological Sciences, University of Utah

[21]School of Forest, Fisheries, and Geomatics Sciences University of Florida, Gainesville, FL,32611, USA

[22]Thünen Institute of Climate-Smart Agriculture, Braunschweig, Germany

[23]Image Processing Laboratory (IPL), Universitat de València, Spain

[24]State Key Laboratory of Vegetation and Environmental Change, Institute of Botany, Chinese Academy of Sciences, Beijing 100093, P. R. China

[25]College of Science and Engineering, James Cook University, Cairns, Queensland, Australia

[26]Dept of Atmospheric and Oceanic Sciences, University of Wisconsin-Madison, Madison, WI 53706 USA

[27]Shanxi University, China

[28]University of Alaska Fairbanks, Institute of Arctic Biology, Fairbanks, AK 99775, USA

[29]Environmental Protection Agency of Aosta Valley, Climate Change Unit, (ARPA Valle d'Aosta), Italy

[30]Dept. of Mathematics and Physics, Catholic University of the Sacred Heart, Brescia (Italy)

[31]Plants and Ecosystems (PLECO), Department of Biology, University of Antwerp, B-2610 Wilrijk, Belgium

[32]USDA-ARS

[33]Virginia Commonwealth University, Department of Biology

[34]Terra Teaching and Research Center, University of Liège – Gembloux Agro-Bio Tech, 5030 Gembloux, Belgium.

[35]Center for Environmental Remote Sensing, Chiba University, Japan

[36]Department of Ecoscience, Aarhus University, Frederiksborgvej 399, Roskilde 4000, Denmark

[37]Department of Physical Geography and Ecosystem Science, Lund University, Sölvegatan 12, 223 62 Lund, Sweden

[38]Bioclimatology, University of Goettingen, 37077 Goettingen, Germany

[39]Department of Geography, McGill University, Montreal, Canada

[40]Department of Geography, The University of British Columbia, Vancouver, Canada

[41]Research Institute for Global Change, Japan Agency for Marine-Earth Science and Technology

[42]Finnish Meteorological Institute, Climate System Research Unit, P.O. Box 503, 00101 Helsinki, Finland

[43]Institute of Atmospheric and Earth System Research / Physics, University of Helsinki, Helsinki, Finland

[44]Institute of Landscape Ecology, University of Münster, Münster, Germany

[45]Catholic University of the Sacred Heart, Dep. of Mathematics and Physics, via Garzetta 48 Brescia, Italy

[46]Environmental Science Division, Argonne National Laboratory, Lemont, IL, USA

[47]University of Chicago Consortium for Advanced Science & Engineering (CASE), Chicago, IL, USA

[48]Northwestern Argonne Institute of Science and Engineering, Evanston, IL, USA

[49]National Ecological Observatory Network, Battelle, 1685 38th Street, 80301 Boulder, CO, USA

[50]Department of Atmospheric and Oceanic Sciences, University of Wisconsin-Madison, 1225 W Dayton St, 53711 Madison, WI, USA

[51]Free University of Bolzano, Faculty of Agricultural, Environmental and Food Sciences - Universitätsplatz 1 - Piazza Università, 1 39100 Bozen-Bolzano

[52]Belle W. Baruch Institute of Coastal Ecology and Forest Science, Clemson University, Georgetown, SC, United States

[53]Forestry and Environmental Conservation Department, Clemson University, Clemson, SC, United States

[54]CEFE, Univ Montpellier, CNRS, EPHE, IRD, Montpellier, France

[55]Department of Forest Ecology and Management, Swedish University of Agricultural Sciences, Skogsmarksgränd 17, SE-901 83, Umeå, Sweden

[56]Hawkesbury Institute for the Environment, Western Sydney University, Penrith Australia 2751

[57]Departamento de Química e Física, Universidade Federal da Paraíba - Campus II, 58397-000 Areia, Paraíba, Brazil

[58]Plants and Ecosystems, Department of Biology, University of Antwerp, 2610 Wilrijk, Belgium

[59]University of Tuscia, Department for Innovation in Biological, Agro-food and Forest Systems (DIBAF)

[60]GFZ German Research Centre for Geosciences, Potsdam, Germany

[61]Forschungszentrum Jülich, Institute of Bio-and Geosciences: Agrosphere (IBG-3), 52428 Jülich, Germany

[62]Woodwell Climate Research Center: FALMOUTH, MA, US

[63]School of Science, Edith Cowan University, Australia

[64]University of Florida, Range Cattle Research and Education Center. 3401 Experiment Station, Ona, FL, USA 33865.



[65]Department of Agriculture Sciences, University of Sassari, Italy

[66]Department of Physical Geography and Ecosystem Science, Lund University, Sölvegatan 12, SE-223 62 Lund, Sweden

[67]Department of Geosciences and Natural Resource Management, University of Copenhagen, Øster Voldgade 10, DK-1350 Copenhagen, Denmark

[68]Terrasystem srl

[69]Institute for Atmospheric and Earth System Research/Physics, Faculty of Science, University of Helsinki, Helsinki, 00014, Finland

[70]Institute for Atmospheric and Earth System Research/Forest Sciences, Faculty of Agriculture and Forestry, University of Helsinki, Helsinki, 00014, Finland

[71]UCLouvain - Erath and Life Institute (ELI) Croix du Sud, 2 bte L7.05.24 (de Serres B249) Ž013 1348 Louvain-la-Neuve

[72]Department of Methods and Models for Economics, Territory and Finance (MEMOTEF), Sapienza University of Rome, Via del Castro Laurenziano, 9, 00161, Rome,Italy

[73]School of Sustainable Engineering and the Built Environment, Arizona State University, Tempe, AZ, USA 85287-8704

[74]Center for Hydrologic Innovations, Arizona State University, Tempe, AZ, USA 85287-8704

[75]School of the Environment, The University of Queensland, 4072, Australia

[76]CSIRO, Space and Astronomy, Kensington, 6151, WA, Australia

[77]Instituto Tecnológico de Sonora, 5 de Febrero 818 Sur, Col. Centro, Cd. Obregon, Sonora, México, 85000

[78]School of life sciences, University of Qufu normal university, Qufu 273165, Shandong, China

[79]Institute of Applied Ecology, Chinese Academy of Sciences, Shenyang 110016, China

[80]Department Biology, San Diego State University, San Diego, CA 92182, USA

**Correspondence:** Jacob A Nelson (jnelson@bgc-jena.mpg.de) and Sophia Walther (sophia.walther@bgc-jena.mpg.de)

**Abstract.** Mapping in-situ eddy covariance measurements of terrestrial land-atmosphere fluxes to the globe is a key method for diagnosing the Earth system from a data-driven perspective. We describe the first global products (called X-BASE) from a newly implemented up-scaling framework, FLUXCOM-X. The X-BASE products comprise of estimates of $CO_2$ net ecosystem exchange ($NEE$), gross primary productivity ($GPP$) as well as evapotranspiration ($ET$) and, for the first time, a novel

fully data-driven global transpiration product ($ET_T$), at high spatial (0.05°) and temporal (hourly) resolution. X-BASE estimates the global $NEE$ at -5.75 ± 0.33 $PgC \cdot yr^{-1}$ for the period 2001-2020, showing a much higher consistency with independent atmospheric carbon cycle constraints compared to the previous versions of FLUXCOM. The improvement of global $NEE$ was likely only possible thanks to the international effort to increase the precision and consistency of eddy covariance collection and processing pipelines, as well as to the extension of the measurements to more site-years resulting in a

wider coverage of bio-climatic conditions. However, X-BASE global net ecosystem exchange shows a very low inter-annual variability, which is common to state-of-the-art data-driven flux products and remains a scientific challenge. With 125 ± 2.1 $PgC \cdot yr^{-1}$ for the same period, X-BASE $GPP$ is slightly higher than previous FLUXCOM estimates, mostly in temperate and boreal areas. X-BASE evapotranspiration amounts to $74.7x10^3 ± 0.9x10^3\ km^3$ globally for the years 2001-2020, but exceeds precipitation in many dry areas likely indicating overestimation in these regions. On average 57% of evapotranspiration

are estimated to be transpiration, in good agreement with isotope-based approaches, but higher than estimates from many land surface models. Despite considerable improvements to the previous up-scaling products, many further opportunities for development exist. Pathways of exploration include methodological choices in the selection and processing of eddy-covariance and satellite observations, their ingestion into the framework, and the configuration of machine learning methods. For this, the new



FLUXCOM-X framework was specifically designed to have the necessary flexibility to experiment, diagnose, and converge to
more accurate global flux estimates.

## 1 Introduction

Energy, water, and carbon exchange between terrestrial surfaces and the atmosphere are key components of the Earth system
and impact ecosystems, ecosystem services, weather, climate, and water availability. The exchange (or flux) can be directly
observed using eddy covariance (EC) measurement systems (Baldocchi, 2019) which are installed on towers overlooking the
ecosystem of interest. The EC stations typically represent an area of a few hundred square meters to a square kilometer.
One key advantage of the EC methodology is the ability to provide near continuous measurements with some records now
exceeding 20 years (Pastorello et al., 2020), allowing for examination of flux variations from the order of thirty minutes to
decades. EC systems also provide a unique perspective on the magnitude, temporal variability, and environmental sensitivity
of ecosystem $CO_2$ uptake, water use, and local climate regulation (Baldocchi, 2019; Musavi et al., 2017; Bao et al., 2022).
However, while many of the most pressing scientific knowledge gaps surrounding the delicate land carbon balance and the water
cycle require spatially and temporally resolved flux patterns at continental to global scales, EC observations are confined to
individual locations in space and limited periods in time (Kumar et al., 2016; Papale et al., 2015). Methodologies to transcend
the gap between local and global scales are needed to ultimately support societal relevant activities of building greenhouse
gas monitoring systems, taking informed climate and land management actions, and verifying the effectiveness of mitigation
strategies (Baldocchi and Penuelas, 2019; Bonan et al., 2011; Novick et al., 2022).

Coordinated and consolidated data collections from EC networks are invaluable for the mapping of in-situ fluxes to regional
and global scales. For example, EC measurements aid both the parameterization (Huang et al., 2021) and the validation (Turner
et al., 2006; Heinsch et al., 2006) of mechanistic models of ecosystem productivity and land surface processes. The latter
generate widely used reference data sets for terrestrial carbon cycle applications (Zhao and Running, 2010; Ukkola et al.,
2022). A complementary approach to modeling terrestrial fluxes at continental and global scales is of empirical nature and
links observations of explanatory variables at the EC stations, particularly meteorological and remote sensing data, to the EC
fluxes via machine learning models. This up-scaling concept does not prescribe any mechanistic formulations and assumes that
the EC observations cover all complexities of ecosystem functioning. Based on a trained machine learning model and globally
gridded input data of the explanatory variables, EC fluxes can be mapped to the global scale.

First implementations of this flux up-scaling concept emerged in the early 2000s. They focused on net ecosystem exchange
of $CO_2$ ($NEE$) and utilized the growing EC networks in Europe (Papale and Valentini, 2003) and North America (Xiao
et al., 2008). The release of the La Thuile Synthesis Dataset of harmonized EC data in 2007, as well as methodological
improvements in the training of the machine learning models (Jung et al., 2009), led to the first global products of terrestrial
$CO_2$ and water fluxes at a monthly time step and in 0.5° grids in 2011 (Jung et al., 2011). While good agreement of flux
estimates derived from complementary process-based models with the up-scaled global gross photosynthetic $CO_2$ uptake
(gross primary productivity, $GPP$) and energy fluxes demonstrated the potential of the approach, important inconsistencies



remained, in particular regarding the globally integrated $NEE$ and its year-to-year variability. (Jung et al., 2020, 2019, 2011; Zscheischler et al., 2017).

In an effort to better understand the uncertainties associated with mapping of EC fluxes to larger scales, the FLUXCOM inter-comparison initiative built an ensemble of flux estimates as a type of factorial experiment (Tramontana et al., 2016; Jung et al., 2019, 2020). The ensemble consisted of multiple machine learning algorithms, meteorological forcing data, and combinations of predictor variables resulting in 120 individual up-scaled estimates per flux. These were summarized in two overall ensemble configurations, which differed in the set of predictors and spatial-temporal resolution. Apart from creating a large ensemble, the FLUXCOM evaluation included a consistent site-level cross-validation analysis as well as cross-consistency checks with terrestrial flux estimates from independent approaches, such as complementary modeling concepts or observational surrogates. From a methodological point of view, the key lessons learned from FLUXCOM were that: (1) the overall approach seems to be primarily limited by the input information given to the machine learning algorithms rather than to the ability of the algorithm to extract the information; (2) the largest qualitative differences among flux products were related to the set of the predictor variables rather than to the choice of the machine learning method or meteorological forcing; (3) the cross-consistency checks with global independent data are essential for supplementing site-level cross-validation; and (4) the largest qualitative discrepancy with independent data was a very high (strongly negative) tropical $NEE$ that was shared among all ensemble members.

By today, the empirical up-scaling concept has been implemented for a series of regional and global scale applications, each of them adopting disparate and individual methodological choices (e.g. Ichii et al., 2017; Yao et al., 2018; Joiner and Yoshida, 2020; Virkkala et al., 2021; Dannenberg et al., 2023; Burton et al., 2023). These potentially important choices relate to data treatment (quality control, gap-filling, processing pathways), ingestion (sampling, as well as matching EC and space-born observations), and methodological configurations (machine learning methods and their training configuration, choice of predictor variables, resolution). Hence, flexibility to explore the large methodological space, as well as the ability to diagnose and evaluate global products in parallel to site-level cross-validation, are required to make progress in empirical up-scaling of EC fluxes. Learning from key insights in FLUXCOM and other up-scaling exercises further implies striving for enhancing the information content of the training data with aspects related to coverage and quality of EC measurements as well as quality, complementarity, and completeness of predictor variables.

We are developing a modeling framework that allows experimenting with and systematically exploring many of these methodological choices. We coin this extended and flexibly adjustable up-scaling framework FLUXCOM-X. Based on FLUXCOM-X, the latency with which innovations in the related fields of machine learning and spacebased Earth observations as well as novel EC data can find their way to empirical flux up-scaling will be considerably reduced. This in turn allows faster progress towards more accurate and fit-for-the-purpose global biogenic flux estimates. Here, we introduce and evaluate the initial "basic" set of products from this framework, which we refer to as FLUXCOM-X-BASE products (or X-BASE for short).

X-BASE products were generated based on the same principle as in the original FLUXCOM ensemble using qualitatively similar predictor variables, i.e. remotely sensed vegetation indices and land surface temperatures from the Moderate Resolution Imaging Spectroradiometer (MODIS) along with meteorological variables. We made efforts to provide more and improved





information to the machine learning models by enhancing coverage and quality of the training data in X-BASE, and by further developing the processing of satellite predictor variables (Walther and Besnard et al., 2022). In this manuscript, we show results for X-BASE $NEE$, $GPP$, evapotranspiration ($ET$), and for the first time transpiration ($ET_T$), for the period 2001-

2020 at 0.05° spatial and hourly temporal resolution. X-BASE products are freely available and serve as a baseline for future FLUXCOM-X developments. We are focusing here on the evaluation and cross-consistency checks of X-BASE with previous FLUXCOM products and independent data streams. Our specific objectives are:

1. to describe the production of X-BASE products;
2. to evaluate the X-BASE setup using site-level cross-validation;

3. to assess qualitative differences of global patterns compared to previous FLUXCOM products with reference to independent flux estimates where possible; and

4. to synthesize lessons learned from this exercise to guide future FLUXCOM-X developments.

## 2    Data and Methods

The following section gives an overview on the essential methodological implementations and data choices adopted in the

generation of X-BASE products.

### 2.1    Eddy Covariance Data

Eddy covariance data consisted of 294 sites from around the world though skewed towards higher representation from temperate forests from North America and Europe. All EC data were collected, processed, analyzed for quality by the station teams, before being processed using state-of-the-art approaches in the ONEFLUX data processing pipeline (Pastorello et al., 2020).

The data included was collected between 2001-2020 and available with a CC BY 4.0 license. Based on this criterion, data for each site came from one of five different sources based on most recent availability: FLUXNET 2015 (Pastorello et al., 2020), ICOS Drought 2018 (Team and Centre, 2020), ICOS Warm Winter 2020 (Team and Centre, 2022), or the most recent Ameriflux or ICOS release as of December 2022. Table 1 lists all sites included as well as the associated digital object identifier specific to the associated release.

Table 1: **Citation data for the 294 sites used in the X-BASE products.**

| AR-SLu(Garcia et al., 2016) | AR-TF1(Kutzbach, 2021) | AR-Vir(Posse et al., 2016) | AT-Neu(Wohlfahrt et al., 2016) | AU-ASM(Cleverly and Eamus, 2016b) | AU-Ade(Beringer and Hutley, 2016c) |
|---|---|---|---|---|---|



Table 1: **Citation data for the 294 sites used in the X-BASE products.**

| | | | | | |
|---|---|---|---|---|---|
| AU-Cpr(Meyer et al., 2016) | AU-Cum(Pendall and Griebel, 2016) | AU-DaP(Beringer and Hutley, 2016b) | AU-DaS(Beringer and Hutley, 2016f) | AU-Dry(Beringer and Hutley, 2016e) | AU-Emr(Schroder et al., 2016) |
| AU-Fog(Beringer and Hutley, 2016a) | AU-Gin(Macfarlane et al., 2016) | AU-RDF(Beringer and Hutley, 2016d) | AU-Rob(Liddell, 2016) | AU-TTE(Cleverly and Eamus, 2016a) | AU-Tum(Woodgate et al., 2016) |
| AU-Wac(Beringer et al., 2016b) | AU-Whr(Beringer et al., 2016a) | AU-Wom(Arndt et al., 2016) | AU-Ync(Beringer and Walker, 2016) | BE-Bra(Team and Centre, 2022) | BE-Dor(Team and Centre, 2022) |
| BE-Lcr(RI, 2021) | BE-Lon(Team and Centre, 2022) | BE-Maa(Team and Centre, 2022) | BE-Vie(Team and Centre, 2022) | BR-Npw(Vourlitis et al., 2022) | BR-Sa1(Saleska, 2016) |
| BR-Sa3(Goulden, 2016d) | CA-Cbo(Staebler, 2022) | CA-DB2(Knox, 2022) | CA-DBB(Christen and Knox, 2022) | CA-ER1(Wagner-Riddle, 2021) | CA-Gro(McCaughey, 2016) |
| CA-LP1(Black, 2021) | CA-Man(Amiro, 2016b) | CA-NS2(Goulden, 2016a) | CA-NS3(Goulden, 2016b) | CA-NS4(Goulden, 2016c) | CA-NS5(Goulden, 2016g) |
| CA-NS6(Goulden, 2016e) | CA-NS7(Goulden, 2016f) | CA-Oas(Black, 2016b) | CA-Obs(Black, 2016a) | CA-Qfo(Margolis, 2016) | CA-SF1(Amiro, 2016c) |
| CA-SF2(Amiro, 2016a) | CA-SF3(Amiro, 2016d) | CA-TP1(Arain, 2016b) | CA-TP2(Arain, 2016a) | CA-TP3(Arain, 2022b) | CA-TP4(Arain, 2016c) |
| CA-TPD(Arain, 2022a) | CG-Tch(Nouvellon, 2016) | CH-Aws(Team and Centre, 2022) | CH-Cha(Team and Centre, 2022) | CH-Dav(Team and Centre, 2022) | CH-Fru(Team and Centre, 2022) |
| CH-Lae(Team and Centre, 2022) | CH-Oe1(Ammann, 2016) | CH-Oe2(Team and Centre, 2022) | CN-Cha(Zhang and Han, 2016) | CN-Cng(Dong, 2016) | CN-Dan(Shi et al., 2016) |
| CN-Din(Zhou and Yan, 2016) | CN-Du2(Chen, 2016k) | CN-Du3(Shao, 2016b) | CN-HaM(Tang et al., 2016) | CN-Qia(Wang and Fu, 2016) | CN-Sw2(Shao, 2016a) |
| CZ-BK1(Team and Centre, 2022) | CZ-BK2(Sigut et al., 2016) | CZ-KrP(Team and Centre, 2022) | CZ-Lnz(Team and Centre, 2022) | CZ-RAJ(Team and Centre, 2022) | CZ-Stn(Team and Centre, 2022) |



Table 1: **Citation data for the 294 sites used in the X-BASE products.**

| | | | | | |
|---|---|---|---|---|---|
| CZ-wet(Team and Centre, 2022) | DE-Akm(Team and Centre, 2022) | DE-Geb(Team and Centre, 2022) | DE-Gri(Team and Centre, 2022) | DE-Hai(Team and Centre, 2022) | DE-HoH(Team and Centre, 2022) |
| DE-Hte(Team and Centre, 2020) | DE-Hzd(Team and Centre, 2022) | DE-Kli(Team and Centre, 2022) | DE-Lkb(Lindauer et al., 2016) | DE-Lnf(Knohl et al., 2016) | DE-Obe(Team and Centre, 2022) |
| DE-RuR(RI, 2022) | DE-RuS(Team and Centre, 2022) | DE-RuW(Team and Centre, 2022) | DE-Seh(Schneider and Schmidt, 2016) | DE-SfN(Klatt et al., 2016) | DE-Spw(Bernhofer et al., 2016) |
| DE-Tha(Team and Centre, 2022) | DE-Zrk(Sachs et al., 2016) | DK-Eng(Pilegaard and Ibrom, 2016) | DK-Fou(Olesen, 2016) | DK-Gds(RI, 2022) | DK-Sor(Team and Centre, 2022) |
| ES-Abr(Team and Centre, 2022) | ES-Agu(Team and Centre, 2022) | ES-Amo(Poveda et al., 2016) | ES-Cnd(Team and Centre, 2022) | ES-LJu(Team and Centre, 2022) | ES-LM1(Team and Centre, 2022) |
| ES-LM2(Team and Centre, 2022) | ES-LgS(Reverter et al., 2016b) | ES-Ln2(Reverter et al., 2016a) | FI-Hyy(Team and Centre, 2022) | FI-Jok(Lohila et al., 2016) | FI-Ken(Team and Centre, 2022) |
| FI-Let(Team and Centre, 2022) | FI-Lom(Aurela et al., 2016a) | FI-Qvd(Team and Centre, 2022) | FI-Sii(Team and Centre, 2022) | FI-Sod(Aurela et al., 2016b) | FI-Var(RI, 2022) |
| FR-Aur(Team and Centre, 2022) | FR-Bil(Team and Centre, 2022) | FR-EM2(RI, 2022) | FR-FBn(Team and Centre, 2022) | FR-Fon(Team and Centre, 2022) | FR-Gri(Team and Centre, 2022) |
| FR-Hes(Team and Centre, 2022) | FR-LBr(Berbigier and Loustau, 2016) | FR-LGt(RI, 2022) | FR-Lam(Team and Centre, 2022) | FR-Pue(Ourcival, 2016) | FR-Tou(RI, 2022) |
| GF-Guy(Team and Centre, 2022) | GH-Ank(Valentini et al., 2016b) | GL-Dsk(RI, 2022) | GL-NuF(Hansen, 2016) | GL-ZaF(Lund et al., 2016b) | GL-ZaH(Lund et al., 2016a) |
| IE-Cra(Team and Centre, 2022) | IL-Yat(Team and Centre, 2022) | IT-BCi(Team and Centre, 2022) | IT-BFt(RI, 2022) | IT-CA1(Sabbatini et al., 2016c) | IT-CA2(Sabbatini et al., 2016a) |
| IT-CA3(Sabbatini et al., 2016b) | IT-Col(Matteucci, 2016) | IT-Cp2(Team and Centre, 2022) | IT-Cpz(Valentini et al., 2016a) | IT-Isp(Gruening et al., 2016b) | IT-La2(Cescatti et al., 2016) |
| IT-Lav(Team and Centre, 2022) | IT-Lsn(RI, 2022) | IT-MBo(Team and Centre, 2022) | IT-Noe(Spano et al., 2016) | IT-PT1(Manca and Goded, 2016) | IT-Ren(Team and Centre, 2022) |
| IT-Ro1(Valentini et al., 2016c) | IT-Ro2(Papale et al., 2016) | IT-SR2(Team and Centre, 2022) | IT-SRo(Gruening et al., 2016a) | IT-Tor(Team and Centre, 2022) | JP-MBF(Kotani, 2016b) |



Table 1: **Citation data for the 294 sites used in the X-BASE products.**

| | | | | | |
|---|---|---|---|---|---|
| JP-SMF(Kotani, 2016a) | MX-Tes(Yepez and Garatuza, 2021) | MY-PSO(Kosugi and Takanashi, 2016) | NL-Hor(Dolman et al., 2016a) | NL-Loo(Team and Centre, 2020) | PA-SPn(Wolf et al., 2016b) |
| PA-SPs(Wolf et al., 2016a) | PE-QFR(Griffis and Roman, 2021) | RU-Che(Merbold et al., 2016) | RU-Cok(Dolman et al., 2016b) | RU-Fy2(Team and Centre, 2022) | RU-Fyo(Team and Centre, 2022) |
| RU-Ha1(Belelli et al., 2016) | SD-Dem(Ardö et al., 2016) | SE-Deg(Team and Centre, 2022) | SE-Htm(Team and Centre, 2022) | SE-Lnn(Team and Centre, 2020) | SE-Nor(Team and Centre, 2022) |
| SE-Ros(Team and Centre, 2022) | SE-Svb(Team and Centre, 2022) | SJ-Adv(Christensen, 2016) | SJ-Blv(Boike et al., 2016) | SN-Dhr(Tagesson et al., 2016) | US-A32(Billesbach et al., 2022) |
| US-AR1(Billesbach et al., 2016b) | US-AR2(Billesbach et al., 2016a) | US-ARM(Biraud et al., 2022) | US-ARb(Torn, 2016b) | US-ARc(Torn, 2016a) | US-Atq(Zona and Oechel, 2016a) |
| US-BZB(Euskirchen, 2022b) | US-BZF(Euskirchen, 2022c) | US-BZS(Euskirchen, 2022d) | US-BZo(Euskirchen, 2022a) | US-Bi1(Rey-Sanchez et al., 2022b) | US-Bi2(Rey-Sanchez et al., 2022a) |
| US-Blo(Goldstein, 2016) | US-CF1(Huggins, 2021) | US-CF2(Huggins, 2022c) | US-CF3(Huggins, 2022a) | US-CF4(Huggins, 2022b) | US-CRT(Chen and Chu, 2016b) |
| US-CS1(Desai, 2022a) | US-CS2(Desai, 2022c) | US-CS3(Desai, 2022d) | US-CS4(Desai, 2022b) | US-Cop(Bowling, 2016) | US-EDN(Oikawa, 2021) |
| US-GBT(Massman, 2016) | US-GLE(Massman, 2022) | US-Goo(Meyers, 2016b) | US-HB1(Forsythe et al., 2021) | US-HWB(Goslee, 2022) | US-Ha1(Munger, 2016) |
| US-Hn3(Liu et al., 2022) | US-Ho2(Hollinger, 2022) | US-IB2(Matamala, 2016) | US-ICs(Euskirchen et al., 2022a) | US-ICt(Euskirchen et al., 2022b) | US-Ivo(Zona and Oechel, 2016b) |
| US-Jo2(Vivoni and Perez-Ruiz, 2022) | US-KFS(Brunsell, 2022a) | US-KLS(Brunsell, 2022b) | US-KS1(Drake and Hinkle, 2016a) | US-KS2(Drake and Hinkle, 2016b) | US-KS3(Hinkle, 2022) |
| US-LWW(Meyers, 2016a) | US-Lin(Fares, 2016) | US-Los(Desai, 2016c) | US-MMS(Novick and Phillips, 2022) | US-MOz(Wood and Gu, 2022) | US-Me1(Law, 2016c) |



Table 1: **Citation data for the 294 sites used in the X-BASE products.**

| | | | | | |
|---|---|---|---|---|---|
| US-Me2(Law, 2022) | US-Me3(Law, 2016a) | US-Me4(Law, 2016e) | US-Me5(Law, 2016d) | US-Me6(Law, 2016b) | US-Mpj(Litvak, 2021) |
| US-Myb(Sturtevant et al., 2016) | US-NGB(Torn and Dengel, 2021) | US-NR1(Blanken et al., 2022) | US-Ne1(Suyker, 2022) | US-Ne2(Suyker, 2016b) | US-Ne3(Suyker, 2016a) |
| US-ONA(Silveira, 2021) | US-ORv(Bohrer, 2021) | US-OWC(Bohrer and Kerns, 2022) | US-Oho(Chen et al., 2016) | US-PFa(Desai, 2016d) | US-Prr(Kobayashi and Suzuki, 2016) |
| US-Rms(Flerchinger, 2022c) | US-Ro1(Baker et al., 2022) | US-Ro4(Baker and Griffis, 2022a) | US-Ro5(Baker and Griffis, 2021) | US-Ro6(Baker and Griffis, 2022b) | US-Rwe(Flerchinger and Reba, 2022) |
| US-Rwf(Flerchinger, 2022a) | US-Rws(Flerchinger, 2022b) | US-SRC(Kurc, 2022) | US-SRG(Scott, 2016a) | US-SRM(Scott, 2016b) | US-Sne(Shortt et al., 2022) |
| US-Snf(Kusak et al., 2022) | US-Sta(Ewers and Pendall, 2016) | US-Syv(Desai, 2016b) | US-Ton(Baldocchi and Ma, 2016) | US-Tw1(Valach et al., 2021) | US-Tw2(Sturtevant et al., 2022) |
| US-Tw3(Chamberlain et al., 2022) | US-Tw4(Sanchez et al., 2016) | US-Tw5(Valach et al., 2022) | US-Twt(Baldocchi, 2016) | US-UM3(Bohrer, 2022) | US-UMB(Gough et al., 2016) |
| US-UMd(Gough et al., 2022) | US-Var(Baldocchi et al., 2016) | US-WCr(Desai, 2016a) | US-WPT(Chen and Chu, 2016a) | US-Whs(Scott, 2016d) | US-Wi0(Chen, 2016g) |
| US-Wi1(Chen, 2016e) | US-Wi2(Chen, 2016j) | US-Wi3(Chen, 2016b) | US-Wi4(Chen, 2016d) | US-Wi5(Chen, 2016a) | US-Wi6(Chen, 2016h) |
| US-Wi7(Chen, 2016i) | US-Wi8(Chen, 2016c) | US-Wi9(Chen, 2016f) | US-Wjs(Litvak, 2022) | US-Wkg(Scott, 2016c) | US-xBR(Network), 2022) |

Meteorological data measured at each site consisted of incoming shortwave radiation, air temperature and vapor pressure deficit, of which all data were gap-filled using the Marginal Distribution Sampling method (Reichstein et al., 2005), as well as the computed potential shortwave incoming radiation (top of atmosphere theoretical maximum radiation) for every hour. Carbon dioxide flux data consisted of gap-filled net ecosystem exchange ($NEE$, variable ustar threshold 50th percentile i.e., NEE_VUT_50) and the corresponding gross primary productivity ($GPP$, nighttime partitioning method (Reichstein et al.,



2005)). Water flux data consisted of evapotranspiration ($ET$, no energy balance correction) which was converted from the latent energy and transpiration estimates based on the Transpiration Estimation Algorithm (TEA) (Nelson et al., 2018; Nelson, 2021). All data were aggregated to a common hourly time resolution, an overview of which can be found in Table 2.

Table 2: **Fluxes to be predicted and predictor variables used in X-BASE.** The units of the fluxes correspond to the native hourly resolution. Upon temporal aggregation as in some analyses in the presented results, the units may change.

| predicted fluxes | | |
| --- | --- | --- |
| $NEE$ | $\mu mol\, CO_2 \cdot m^{-2} \cdot s^{-1}$ | net ecosystem exchange |
| $GPP$ | $\mu mol\, CO_2 \cdot m^{-2} \cdot s^{-1}$ | gross primary productivity |
| $ET$ | $mm \cdot hr^{-1}$ | evapotranspiration |
| $ET_T$ | $mm \cdot hr^{-1}$ | transpiration |

| predictor variables | |
| --- | --- |
| air temperature | $^{\circ}C$ |
| vapor pressure deficit | $hPa$ |
| incoming shortwave radiation | $W \cdot m^{-2}$ |
| potential incoming shortwave radiation | $W \cdot m^{-2}$ |
| derivative of daily pot. incoming shortwave radiation | $W \cdot m^{-2} \cdot d^{-1}$ |
| derivative of hourly pot. incoming shortwave radiation | $W \cdot m^{-2} \cdot hr^{-1}$ |
| daytime land surface temperature from MODIS TERRA | kelvin |
| nighttime land surface temperature from MODIS TERRA | kelvin |
| enhanced vegetation index | - |
| near-infrared reflectance of vegetation | - |
| normalized difference water index | - |
| plant functional type | - |

Data from the EC dataset that ultimately were used for training the models varied between ~12-14 million site-hours depending on the target variable (i.e. $GPP$, $NEE$, $ET$, or $ET_T$). Training of the machine learning algorithms was only conducted

on hours where all input variables passed quality control. The quality control procedure consisted of two levels, with the first being each hour must have at least one value of good quality measured or gap-filled with confidence (i.e. at least one half hour was either 0 or 1 based on the OneFLUX _QC flags). Second, a set of consistency tests were performed on each used variable to check the consistency both among variables and across sites. As the consistency flags were based on daily aggregates of the meteorological and flux data, entire days were removed if the test indicated inconsistencies among related variables. The



consistency flag also checked the relationship between variables across sites, ensuring that the relationships found across the data are coherent. A detailed explanation of these consistency flags can be found in Jung et al. (2023).

## 2.2  Global Meteorology

For the generation of global flux maps we used hourly meteorological data from ERA5 global reanalysis products at 0.25°
(Hersbach et al., 2020). Variables included air temperature at 2m height, incoming shortwave radiation at the surface, as well

as vapour pressure deficit (computed from relative humidity, air temperature, and surface pressure). Units were converted to correspond to the site level measurements which were used for training the machine learning model, and the data were re-gridded to a 0.05° resolution using bilinear interpolation for every hour.

## 2.3  Satellite Earth Observation

The X-BASE products are based on measurements of the MODerate Imaging Spectroradiometer (MODIS) of surface re-

flectance and land surface temperature from collection v006 at daily resolution. Missing records were gap-filled consistently in both the average time series per EC station and in the global gridded data following the procedures of the FluxnetEO data version 2 (Walther and Besnard et al., 2022; Walther, 2023).

### 2.3.1  Spectral vegetation indices

At site level we used surface reflectance in the first seven MODIS spectral bands from the MCD43A4 v006 reflectance data

set (500 m and daily, where each daily value is inverted from all valid observations within a 16-day window (Schaaf and Wang, 2015b)). The spectral vegetation indices computed from the reflectance data were the enhanced vegetation index (EVI) (Huete et al., 2002), the spectral reflectance of vegetation in the near-infrared (NIRv) (Badgley et al., 2017), and the normalized difference water index (NDWI) with MODIS band 7 as reference (Gao, 1996). We followed the procedure of the FluxnetEO data sets version 2 (Walther and Besnard et al., 2022) for data acquisition from Google Earth Engine for all pixels in a cutout

of 4x4 km² around each EC station, as well as for quality checks in terms of snow cover, land cover, index values outside the defined ranges, and outliers. An iterative approach then determined both, the strictness of the inversion quality of the bidirectional reflectance distribution function (BRDF, based on the MCD43A2 data, (Schaaf and Wang, 2015a)) and the set of pixels in a cutout that shall represent a given EC station. Supporting information section A1 outlines all technical details of the dynamic procedure.

Global data of BRDF-corrected surface reflectance stem from the MCD43C4 v006 data (Schaaf and Wang, 2015b), available in a climate modelling grid of 0.05° with the same temporal sampling and subject to the same removal of snow and water pixels and outlier values like at site level. The BRDF quality control of the global data followed the same dynamic approach (see supporting information A1), which maximized data availability especially in tropical regions.



### 2.3.2 Land surface temperature

Satellite observations of land surface temperature (LST) were based on the MODIS v006 TERRA observations which are available every day at 1 km resolution (Wan et al., 2015). We selected the 1 km$^2$ pixel containing a specific tower and treated the two MODIS LST data streams as independent predictor variables which represent clear-sky LST at a specific time of the day (namely around 10.30 AM and PM local time). Quality checks and gap-filling followed the procedure described in FluxnetEO version 2 (Walther and Besnard et al., 2022).

For the global spatialization of the flux estimates we relied on climate modelling grid LST from the MODIS TERRA data sets (Wan et al., 2015) and apply consistent quality control and imputation of missing values like at site-level.

### 2.3.3 Land cover

Land cover information used the IGBP global vegetation classification. Site level classification was as reported by the principal investigators. Global data were based on the yearly-resolved MODIS MCD12Q1 v006 product (Friedl and Sulla-Menashe,
2019). In order to ease the transition between site and global land cover classifications, an intermediate classification scheme was utilized which translated each classification into characteristics (e.g. trees, crops, needleleaf, deciduous, etc. . . ) based on whether the classification has (value=1.0), might have (value=0.5), does not have (value=0.0) a specific feature, or is unknown (value=-1.0). A full description of this intermediate classification system can be found in supplementary section A2.

### 2.4 Machine Learning Method

All X-BASE products are based on gradient boosted regression trees using the XGBoost library (Chen and Guestrin, 2016). XGBoost is known as a robust algorithm that is able to handle a variety of variable types (numeric, boolean, categorical). Training was conducted using a two-thirds training sub-sampling ratio and a 0.05 learning rate. Boosting was stopped when no model improvement (based on mean squared error of validation data) was observed for ten consecutive rounds, and the best performing model was stored to generate predictions. In all cases, the model reached the stopping criteria relatively quickly,
with the final number of boosting rounds between 80-230, depending on the flux.

### 2.5 Cross-validation

All cross-validation was performed using a 10 fold, leave-site-fold-out scheme, where each fold was constructed by randomly assigning each site to a fold. For each round of cross-validation, eight folds were used for training, one for validation and the remaining one as the test fold for which the actual predictions were made. The leave-site-fold-out scheme ensures that no data
from the sites in the test fold were ever seen by the algorithm during training, and in turn iterated such that each site was in the test set once. As eddy covariance sites are sometimes clustered in the same location (e.g. as different treatments) and can therefore be both physically closely located and not truly independent, sites are assigned to the same fold if they are less than 0.05° apart to reduce over-fitting. We evaluate the accuracy of the cross-validation models by computing the Nash-Sutcliffe modeling efficiency (NSE, Nash and Sutcliffe (1970)), where a negative NSE indicates a model accuracy that is worse than a



mean prediction, while a value close to one indicates high model accuracy. We compute the NSE for each site and for a range
of temporal scales from hourly to inter-annual.

## 2.6  Up-scaling

The final step to train a model to use in the final global prediction step was identical to the training in the cross-validation,
with the exception that, because no test fold was required, we used nine of the ten folds for the training and validation was
done on the remaining fold. The final trained models (one trained model for each target flux) were then used to predict fluxes
at the global scales using the associated globally gridded input variables that correspond to those used at site level, as outlined
in Table 2.

## 2.7  Previous FLUXCOM and independent global flux estimates

We compare X-BASE with up-scaling results from FLUXCOM (Jung et al., 2019, 2020). As mentioned earlier, FLUXCOM
comprised an ensemble of up-scaling experiments that differed in the choice of machine learning method, meteorological forc-
ing data, and which were summarized in two groups of set-ups that shared the same predictor variables and spatiotemporal res-
olution: The "remote-sensing-only" set-up (RS) mostly used spaceborne observations of MODIS as explanatory variables and
produced flux estimates every 8 days at 0.083° resolution, while the 'remote-sensing plus meteorology set-up' (RS+METEO)
produced daily flux estimates at half degree resolution from meteorological predictor variables and an average seasonal cycle
of satellite observations (Tramontana et al., 2016; Jung et al., 2019, 2020). Comparisons to FLUXCOM RS+METEO datasets
always refer to the ensemble over multiple machine learning methods for all realizations driven by the ERA5 meteorology
(Hersbach et al., 2020). RS+METEO uses average seasonal cycles of MODIS v005 observations. For the FLUXCOM RS set-
up we use the ensemble over all machine learning methods. Please note that both the previous RS runs and the X-BASE runs
presented here are driven by data from MODIS v006, but the processing has changed in some aspects such as quality control
and gap-filling.

For evaluating X-BASE $NEE$ globally, in particular its seasonal cycle and for different regions, we used two different at-
mospheric inversion model products: the Orbiting Carbon Observatory-2 (OCO-2) v10 model intercomparison project (Byrne
et al., 2023) and the CarboScope inversion (Rödenbeck et al., 2018) version s99oc_v2022 (Roedenbeck and Heimann, 2022).
Estimates from the OCO-2 came from the the LNLGIS experiment which combines satellite-based column-averaged $CO_2$
(XCO2) retrievals and in-situ $CO_2$ measurements as observational constraints in the assimilation, and consists of 13 different
ensemble members covering the period 2015-2020 with a monthly frequency and 1° spatial resolution (https://gml.noaa.gov/
ccgg/OCO2_v10mip/index.php). The CarboScope product consisted of a single inversion output at the same spatial resolution
as OCO-2, but a longer temporal period from 2001 to 2020. In each case, as the inversion products estimate net biome ex-
change, we subtracted from the inversions data fire emissions as estimated by the Global Fire Emissions Database, Version 4.1
(Randerson et al., 2017).

We compared temporal patterns of X-BASE $GPP$ with the patterns in global retrievals of sun-induced chlorophyll fluores-
cence (SIF) from the Sentinel-5P TROPOMI instrument (Köhler et al., 2018), which under most conditions approximate the





variability in $GPP$. For the comparison we used estimates of daily mean SIF applying a correction factor to instantaneous observations (Zhang et al., 2018) and averaged both X-BASE $GPP$ and TROPOMI SIF to a temporal resolution of 16 days
and 0.5° spatial grids for the common period 04/2018-12/2020.

X-BASE $ET$ and $ET_T$ were cross-compared with transpiration estimates from the Global Land Evaporation Amsterdam Model (GLEAM) v3.6a (Martens et al., 2017; Miralles et al., 2011). GLEAM also utilizes satellite and reanalysis data sets but in a more physically constrained way, relying on semi-empirical models such as the Priestley and Taylor (Priestley and Taylor, 1972) and Gash models (Gash, 1979). Further comparisons were made to precipitation data from GPCC (Schneider
et al., 2022).

## 3 Results

### 3.1 Cross-validation and data space

One important innovation in FLUXCOM-X compared to the previous FLUXCOM ensemble was the training data base, which was larger due to an incease in both number of sites and years. Furthermore, the EC methodology has changed considerably
in many aspects ranging from collection and processing to quality filtering in the last 15 years. We show here one illustrative example of the changes in the environmental space that is represented in the training samples for daily $NEE$: between daily VPD and daily incoming shortwave radiation the distribution of training samples was considerably broader in X-BASE compared to the RS+METEO ensemble (Fig. 1). Furthermore, the number of unique sites contributing to a certain VPD-radiation bin has increased (Fig. B1), i.e. the number of ecosystems sampled in each climatic condition has also increased. The increases
were seen particularly at the margins of the distribution, i.e. for days with high VPD along the full radiation spectrum, and vice versa for days with high radiation conditions along the full VPD spectrum. Remarkably, the number of sites contributing training samples for high VPD *and* high radiation were observed much more frequently (Fig. 1) and at more sites (Fig. B1) compared to RS+METEO - providing more and more varied information for dry conditions.

The results from the ten-fold cross validation showed an overall high performance with most fluxes and scales of variability
having an NSE above 0.6 (Fig. 2). In terms of scales of variability across all fluxes, the monthly mean diel cycle ("diel") and the daily median seasonal cycle ("seasonal") were very regular patterns that the trained models reproduced best. Also, among-site changes ("spatial", except for $NEE$) and monthly aggregated fluxes ("monthly") were reliably predicted. Deviations from the median daily seasonality ("anom") were only moderately reliable with NSE between 0.25 and 0.5. The XGBoost models did not succeed in accurately reproducing inter-annual changes ("i.a.v.") of all fluxes and between-site patterns in $NEE$. Consistently
across all scales, the net fluxes which are directly calculated (i.e., $ET$ and even more so $NEE$) showed lower performance than their respective modelled gross fluxes (i.e., $GPP$ and $ET_T$). Note that the cross validation results from Fig. 2 cannot be quantitatively compared to previous cross validation results in FLUXCOM as the training data are not the same. However, qualitatively the accuracy gradient among fluxes as well as along scales of variability corresponded to patterns identified in FLUXCOM and in comparable empirical modeling activities (Jung et al., 2011; Tramontana et al., 2016; Virkkala et al., 2021;
Dannenberg et al., 2023).



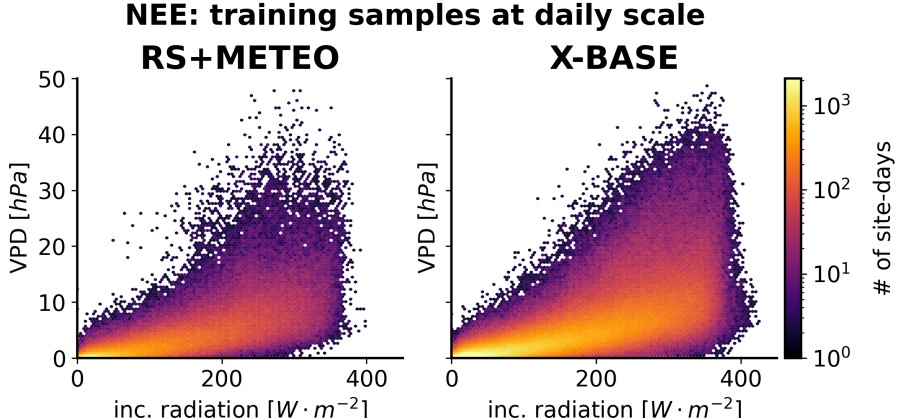

**Figure 1. Cross-validation sampling in meteorological space:** Number of site-days contributing to sampling for $NEE$ for the previous FLUXCOM RS+METEO ensemble (left) compared to the sampling of FLUXCOM-X-BASE (right) in environmental space of daily aggregated incoming shortwave radiation and VPD. Color corresponds to number of site days per bin in log scale. Only bins with at least twenty site-days are shown.

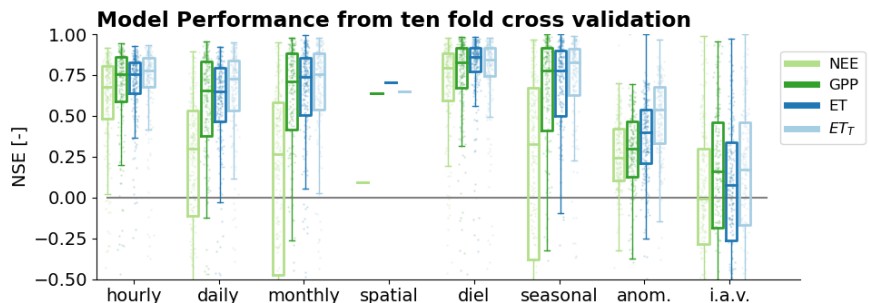

**Figure 2. FLUXCOM-X-BASE site-level accuracy of predicted fluxes** in 10-fold leave-site fold-out cross-validation in terms of NSE computed per site for a range of scales of variability. Scales of variability include the hourly timescale ("hourly"), daily ("daily") and monthly ("monthly") aggregated fluxes, as well as between-site changes ("spatial"), monthly mean diel cycle ("diel"), daily median seasonal cycle ("seasonal"), deviations from the median daily seasonality ("anom."), and inter-annual variability ("i.a.v."). Boxes denote the range from the 25th to the 75th percentile of sites, whiskers extend 1.5 times the interquartile range from the 25th and 75th percentile of NSE across sites.



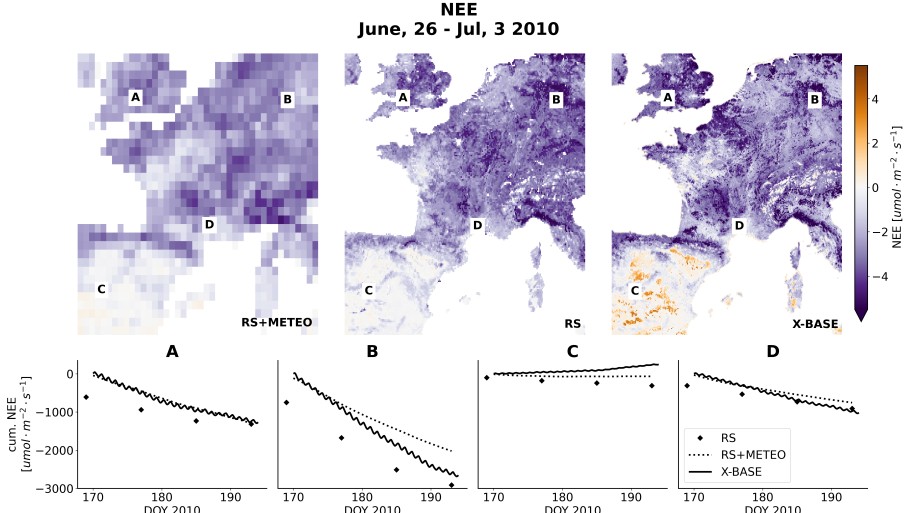

**Figure 3. Resolution improvements for the X-BASE products compared to RS and RS+METEO:** Average $NEE$ for an 8-day period in Europe in 2010 as estimated from the RS, RS+METEO and X-BASE set-ups (top panel), as well as snapshots of temporal trajectories of $NEE$ in pixels closest to selected EC station locations (A:UK-Tad, B: DE-Hai, C: ES-LM1, D: FR-Pue). Negative values of $NEE$ denote a $CO_2$ flux from the atmosphere to the land.

## 3.2 Global flux estimates

One asset of FLUXCOM-X is flexibility in the spatiotemporal resolution of the flux estimates. We are producing X-BASE products at 0.05° spatial and hourly temporal resolution globally. Figure 3 illustrates the increase in spatial and temporal detail in X-BASE compared to RS (0.083°, 8-daily) and RS+METEO (0.5°, daily) using the example of $NEE$.

### 3.2.1 Net Ecosystem Exchange ($NEE$)

The X-BASE product estimates the global terrestrial $NEE$ to be -5.75 ± 0.33 $PgC \cdot yr^{-1}$ (2001-2020), with strong $CO_2$ uptake hotpots in the tropical regions, and temperate regions of North America and Europe (Fig. 4). In contrast to both RS and RS+METEO, India and some regions in central Sahel show prominent patterns of a mean $CO_2$ flux from the ecosystems to the atmosphere in X-BASE, corresponding mostly to crop designated areas (Fig. B2). However, comparing X-BASE global terrestrial $NEE$ to the inversion estimates (corrected for fire emissions based on GFED 4.1 (Randerson et al., 2017)) over the common period (2015-2020) shows agreement of X-BASE (-5.63 $PgC \cdot yr^{-1}$) with OCO-2 (-4.12 $PgC \cdot yr^{-1}$) and CarboScope (-3.46 $PgC \cdot yr^{-1}$).

Comparison with OCO-2 and CarboScope inversions also indicates a substantial improvement of the global mean seasonal cycle of $NEE$ (Fig. 5) in X-BASE compared to RS and RS+METEO. The systematic bias present in RS and RS+METEO has essentially disappeared in X-BASE. The shape, and in particular the amplitude, of the global $NEE$ seasonal cycle of X-BASE





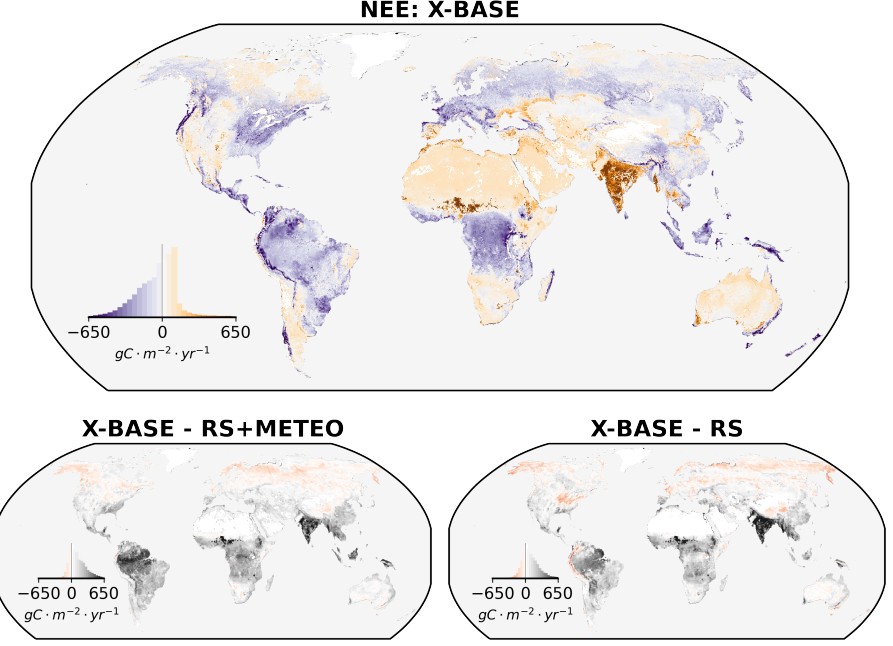

**Figure 4. Comparison of annually integrated** $NEE$ **from X-BASE, RS+METEO with ERA5 forcing and RS** averaged over the period 2001-2020. The difference maps show the difference of the averages over 2001-2020.

is more consistent with the inversions. The larger and more realistic seasonal cycle amplitude of global $NEE$ in X-BASE originates primarily from improved and increased amplitudes in boreal regions. Interestingly, X-BASE suggests slightly larger $NEE$ seasonal cycle amplitudes in temperate regions compared to the inversions. In seasonally dry regions, the timing of maximum uptake is consistent between X-BASE and inversions, while the peak of maximum net release is larger and delayed

in the inversions. In Australia, the peak of $CO_2$ release to the atmosphere at the end of the year present in both inversions is not evident in X-BASE, which instead shows a relatively consistent $CO_2$ flux to the atmosphere throughout the year. In tropical regions, the patterns of seasonal variations are qualitatively consistent between X-BASE and the previous RS and RS+METEO products. The seasonal patterns in tropical regions are relatively weak overall and seem inconsistent both between the inversions and X-BASE as well as among the inversions.

As seen in Figure 5, the X-BASE product shows the same large underestimation of globally integrated $NEE$ inter-annual variance as the previous RS and RS+METEO products. In terms of temporal trends, the X-BASE products show almost no change in annual $NEE$ in time, which is in contrast to the RS+METEO (slight positive trend) and RS (slight negative trend) and more consistent with the CarboScope inversions (Table B2). However, as inter-annual variability was poorly reproduced even in the cross validation (Fig. 2), trends in the X-BASE products should be taken with caution and interpreted with careful

scrutiny.



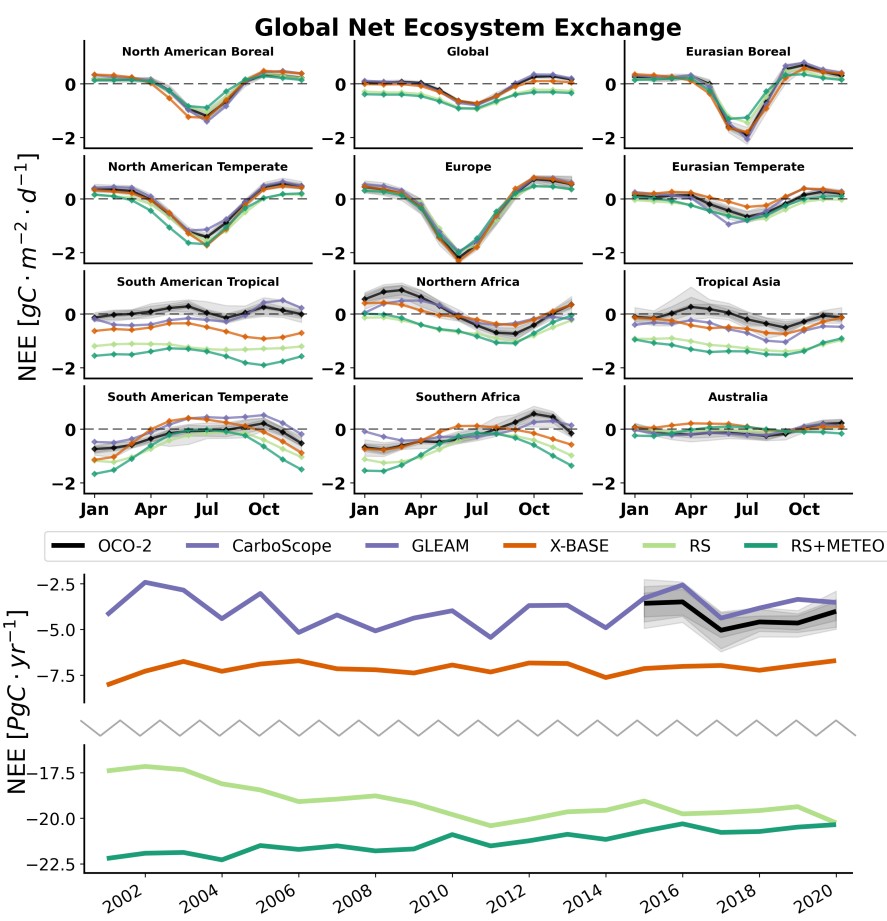

**Figure 5. Seasonal and inter-annual variability of global** $NEE$. Comparison of mean seasonal cycles (calculated over the common time period, 2015-2020) and inter-annual variability (2001-2020) of $NEE$ estimated from CARBOSCOPE and OCO2 inversions as well as FLUXOM-X-BASE and FLUXCOM RS+METEO and RS outputs. All products were integrated with a common mask that removes sparsely vegetated arid regions not predicted by RS and RS+METEO.





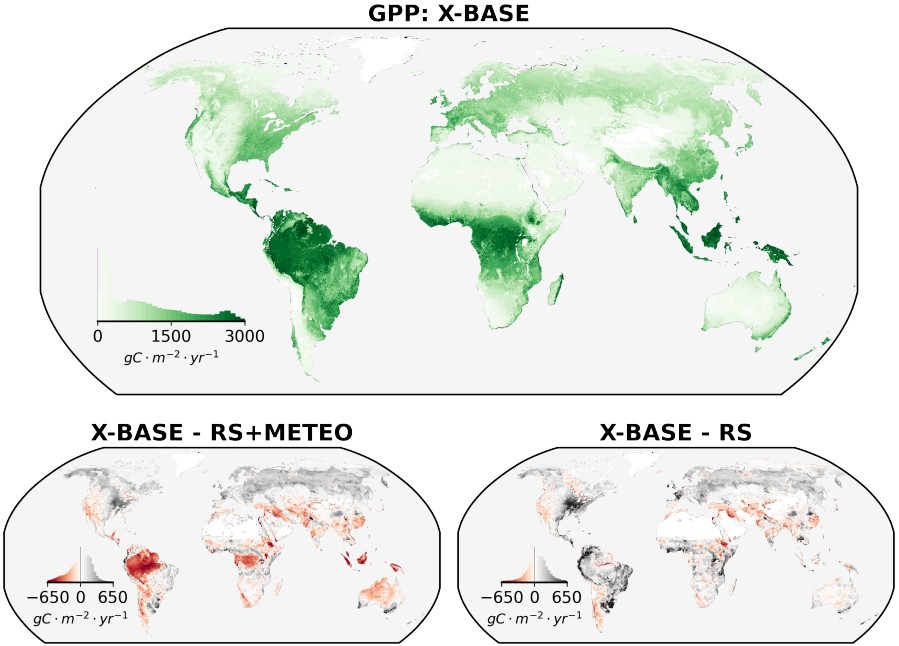

**Figure 6.** Comparison of annually integrated $GPP$ from X-BASE, RS+METEO with ERA5 forcing and RS averaged over the period 2001-2020. The difference maps show the difference of the averages over 2001-2020.

### 3.2.2 Gross Primary Productivity ($GPP$)

X-BASE estimates the globally integrated $GPP$ at $124.7 \pm 2.1\ PgC \cdot yr^{-1}$ on average in the time period 2001-2020. Globally integrated $GPP$ over vegetated areas (RS and RS+METEO do not have estimates for non-vegetated areas) was approximately equal for X-BASE ($121.9 \pm 2.1\ PgC \cdot yr^{-1}$) and RS+METEO ($121.6 \pm 0.4\ PgC \cdot yr^{-1}$) but considerably higher than RS

($113.2 \pm 1.8\ PgC \cdot yr^{-1}$) over the same period. In terms of regional patterns, X-BASE $GPP$ consistently exceeds both RS+METEO and RS in temperate, boreal, and most subtropical ecosystems, but is lower in sparsely vegetated (semi-)arid regions like southwestern North America as well as southeast Asian croplands (Fig. 6). This qualitatively consistent pattern is only broken in the humid tropics, where X-BASE $GPP$ is higher than RS, but lower than RS+METEO.

Comparing the estimated trend over the last two decades, X-BASE $GPP$ has a clear increasing linear trend of $0.34\ PgC \cdot$

$yr^{-2}$ which is slightly higher than the trend in RS ($0.25\ PgC \cdot yr^{-2}$, Table B2). In contrast, the RS+METEO product shows nearly no trend in annual $GPP$. The increases in both the X-BASE and RS products may be related to increases in surface greenness coming from variability in the remote sensing forcing data which are inter-annually dynamic in both products, whereas the remote sensing data were not inter-annually dynamic in the RS+METEO product which instead used only the mean seasonal cycle of the remote sensing data. The magnitude of between-year changes in globally integrated X-BASE $GPP$

is $0.575\ PgC \cdot yr^{-1}$ over the years 2001-2020, which is about twice as large as RS+METEO ($0.248\ PgC \cdot yr^{-1}$), but only half the magnitude estimates in the RS set-up ($1.02\ PgC \cdot yr^{-1}$, Table B2).



We further compared the temporal trajectory in $GPP$ estimates against TROPOMI SIF as an independent proxy for $GPP$ dynamics (Fig. 7) at a temporal resolution of 16 days. The temporal variability of X-BASE $GPP$ strongly agrees with that in TROPOMI SIF, with Squared Spearman correlation values (denoted as $R^2$) of the time series above 0.85 across most of the vegetated land surface (Fig. 7 top left). The only exceptions are regions with no or very small variability in both $GPP$ and SIF such as in either evergreen tropical ecosystems in South America, Africa and southeast Asia, or sparsely to non-vegetated areas due to aridity (e.g. Mexican, and African deserts) or cold conditions (e.g. Canadian and Siberian subpolar regions). In inner Australia, despite being sparsely vegetated, variability between years is expected in $GPP$ due to precipitation increases during La Nina years, which is however not reflected in the squared correlations. $R^2$ for the deviations from the average seasonality (again computed with a temporal resolution of 16 days) show the same qualitative spatial patterns (Fig. 7 top right), but are overall lower with $R^2$ values between 0.55 and 0.8. Anomalies of X-BASE $GPP$ and SIF agree best in eastern European temperate forests as well as grassy and shrub ecosystems in eastern South America.

Comparison of the level of agreement of SIF and X-BASE with that of SIF and RS and RS+METEO, respectively, illustrates that X-BASE and RS $GPP$ estimates have comparable consistency both for the time series (global area weighted mean $R^2$ values of 0.72 and 0.73, respectively) and anomalies (global mean $R^2$ values of 0.64 and 0.66, respectively). In contrast, the $R^2$ between RS+METEO and SIF is lower in both cases ($R^2$ values of 0.66 for the time series and 0.58 for anomalies). X-BASE $GPP$ shows a higher agreement with SIF than RS both in terms of the actual trajectory and anomalies in evergreen tropical forests with no or only a very short dry season in the Amazon and Africa, as well as in fully humid parts of southeast Asia (Fig. 7 middle panel). Improvements in X-BASE $GPP$ compared to RS are also consistent in the very continental and polar tundra areas in eastern Siberia, northern Canada and Alaska. Conversely, in arid steppe climates globally, X-BASE $GPP$ variability agrees less with SIF than does RS $GPP$. X-BASE $GPP$ variability is consistently and widespread much more similar to the variability in TROPOMI SIF than RS-METEO $GPP$. Increases in $R^2$ for X-BASE compared to RS+METEO are most pronounced in arid to semi-arid ecosystems (large parts of the Caatinga and Gran Chaco regions on South America, steppe regions in Mexico, southern and eastern Africa, Australia and central Siberia) as well as in global crop regions, especially for the deviations from the seasonality (albeit the magnitude of $R^2$ change is quite variable between regions, Fig. 7 bottom).

### 3.2.3 Water Vapor Fluxes

Globally integrated $ET$ amounts to $74.7\text{x}10^3 \pm 0.9\text{x}10^3 \ km^3 \cdot yr^{-1}$ for 2001-2020 (Table B1) for X-BASE, with the highest rates in the tropics (Fig. 8). Comparing global totals for vegetated areas only (where all products give outputs) shows similar values for X-BASE ($68.9\text{x}10^3 \pm 0.9\text{x}10^3 \ km^3 \cdot yr^{-1}$), GLEAM ($70.9\text{x}10^3 \pm 0.9\text{x}10^3 \ km^3 \cdot yr^{-1}$) and RS+METEO ($68.3\text{x}10^3 \pm 0.3\text{x}10^3 \ km^3 \cdot yr^{-1}$) $ET$ estimates, while the RS $ET$ is more than 11% higher ($78.5\text{x}10^3 \pm 0.5\text{x}10^3 \ km^3 \cdot yr^{-1}$, Table B1). Particularly in evergreen tropical ecosystems, X-BASE estimates a considerably lower $ET$ than both GLEAM, RS+METEO, and RS (Fig. 8). Furthermore, in the temperate and high latitudes of the northern hemisphere, annually integrated X-BASE $ET$ is consistently lower than the other estimates, though the magnitude of the bias is smaller than in the tropical regions. The pattern is only reversed with higher X-BASE $ET$ in the semi-arid and arid ecosystems of the lower and middle latitudes, especially with respect to annual $ET$ in RS+METEO and GLEAM.



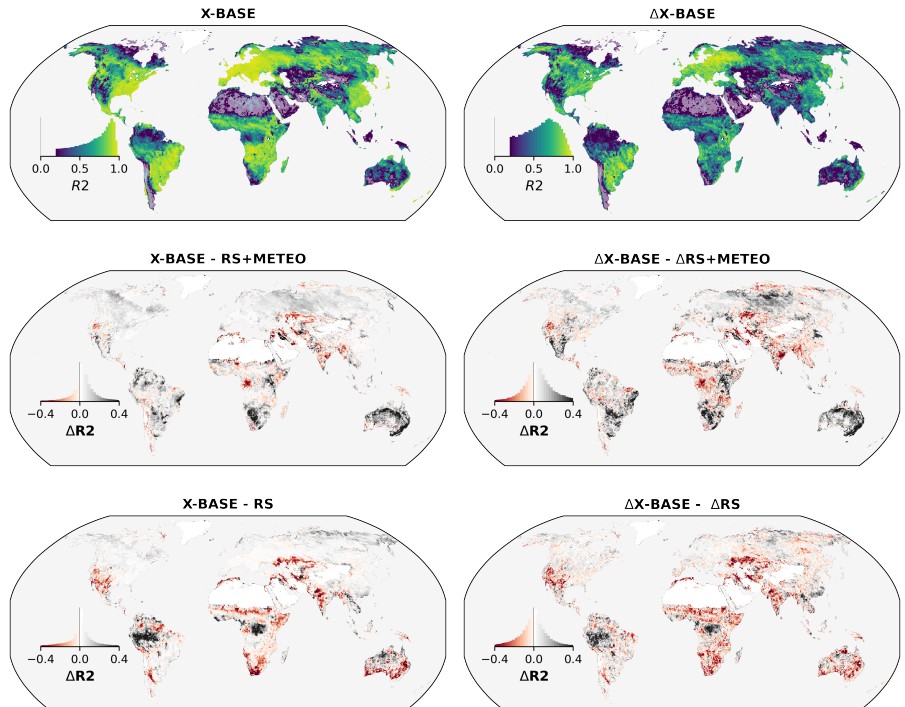

**Figure 7. Similarity of temporal patterns between** $GPP$ **estimates and TROPOMI SIF observations:** $R^2$ (computed as the square of the Spearman correlation) between X-BASE $GPP$ and TROPOMI SIF (Köhler et al., 2018) for the actual time series at a temporal resolution of 16 days (top left) and anomalies from the median seasonality in both variables (top right). The middle and the bottom panels relate the agreement between X-BASE $GPP$ and TROPOMI SIF to the agreement between FLUXCOM $GPP$ and TROPOMI SIF, where the middle panel refers to TROPOMI SIF and $GPP$ from the RS+METEO set-up, and the bottom panel to the RS set-up. All comparisons are done for time series with a resolution of 16 days for the common time period 04/2018 to 12/2020. SIF observations have been applied a correction factor to estimate daily average SIF before aggregation. Semi-transparent areas mark pixels in which the correlation of at least one of the data sets is negative.





Comparison to precipitation estimates shows that X-BASE $ET$ greatly exceeds precipitation inputs over large areas, indicating a strong overestimation of X-BASE $ET$ in many arid regions with sparse vegetation (e.g. the Sahara region, Fig. B3). While transport of water both laterally and from deeper groundwater could cause $ET$ to exceed precipitation inputs in some areas, the extent of area where $ET$ exceeds precipitation (e.g. the entire Sahara region) and the magnitude of the excess $ET$

(over three times precipitation inputs) indicates a major bias in these areas and is likely due to a lack of EC data in similar ecosystems. As a rough estimate, constraining the X-BASE estimates with precipitation (see supplement section B5) suggests about 4-6x$10^3$ $km^3 \cdot yr^{-1}$ of water is overestimated globally.

The globally integrated $ET_T$ amounts to 42.6x$10^3 \pm 1.0$x$10^3$ $km^3 \cdot yr^{-1}$ (2001-2020) in X-BASE, resulting in an average ratio of transpiration to total evaporation of 57.0% $\pm$ 0.6% (Table B1). In contrast to $ET$, the $ET_T$ estimates from X-BASE do

not commonly exceed precipitation estimates (Fig. B3), which could indicate that because the water vapor flux is more tightly coupled with vegetation, the model is able to distinguish that no vegetation corresponds with no transpiration, which is not generally the case for non-transpiration evaporation. The RS and RS+METEO products did not produce $ET_T$ estimates, so the comparison is limited to GLEAM (50.7x$10^3 \pm 0.6$x$10^3$ $km^3 \cdot yr^{-1}$), which estimates $ET_T$ on average 17% higher than X-BASE, with strong contributions from the evergreen tropics. Only in single semi-arid regions, such as northernmost Sahel

as well as large parts of the South American Caatinga and Chaco regions is this pattern reversed (Fig. 8).

Spatially, X-BASE-estimated $ET_T/ET$ exceeds 50% in the majority of areas, with the highest values seen in the higher latitude regions of Europe and Asia, as well as in subtropical ecosystems (Fig. 8). Arid regions with sparse vegetation show the lowest $ET_T/ET$ overall, with values generally below 20%. With 71.4% $\pm$ 0.6% over global vegetated surfaces, GLEAM attributes about 10% more of its $ET$ to $ET_T$ than does X-BASE (Table B1). Regionally, this difference can even reach up to

40%, with the only exception being boreal forests and very dry ecosystems in the Sahel, the Arabian Peninsula and central Asia (Fig. 8).

Trends in $ET$, $ET_T$ and $ET_T/ET$ are positive and exceed the trends seen in all other estimates over the years 2001-2020. Conversely, the magnitude of inter-annual changes in X-BASE $ET$, $ET_T$, and $ET_T/ET$ is mostly less than half than the variability in GLEAM (Table B2). Low inter-annual changes are common to the RS and RS+METEO $ET$ as well.

Figure 9 shows the temporal correlation at 16-daily temporal scale using GLEAM as a reference, showing overall high values of squared correlation between X-BASE and GLEAM $ET$ and $ET_T$ (top and bottom left). Notable exceptions with low correlations are areas with low variability in $ET$ such as the arc of deforestation, very dry areas, and tropical evergreen ecosystems in Africa. Compared to RS+METEO and RS (middle panels left column in Fig. 9), X-BASE $ET$ temporal patterns are more similar to GLEAM $ET$ in many areas, and especially so in areas north of the arc of deforestation and parts of

tropical evergreen areas in central Africa and souteast Asia. Conversely, X-BASE $ET$ agrees less well with GLEAM than RS or RS+METEO in the arc of deforestation itself, the eastern parts of the Amazon basin, as well as dry areas. The deviations from the mean annual cycle in $ET$ and $ET_T$ (right column) show overall lower correlations than the actual time series, with the highest agreement between GLEAM and X-BASE in large parts of the Amazon forest and central European ecosystems. X-BASE $ET$ anomalies are much more strongly correlated with GLEAM $ET$ than either RS or RS+METEO everywhere

except for most (semi-)arid regions.



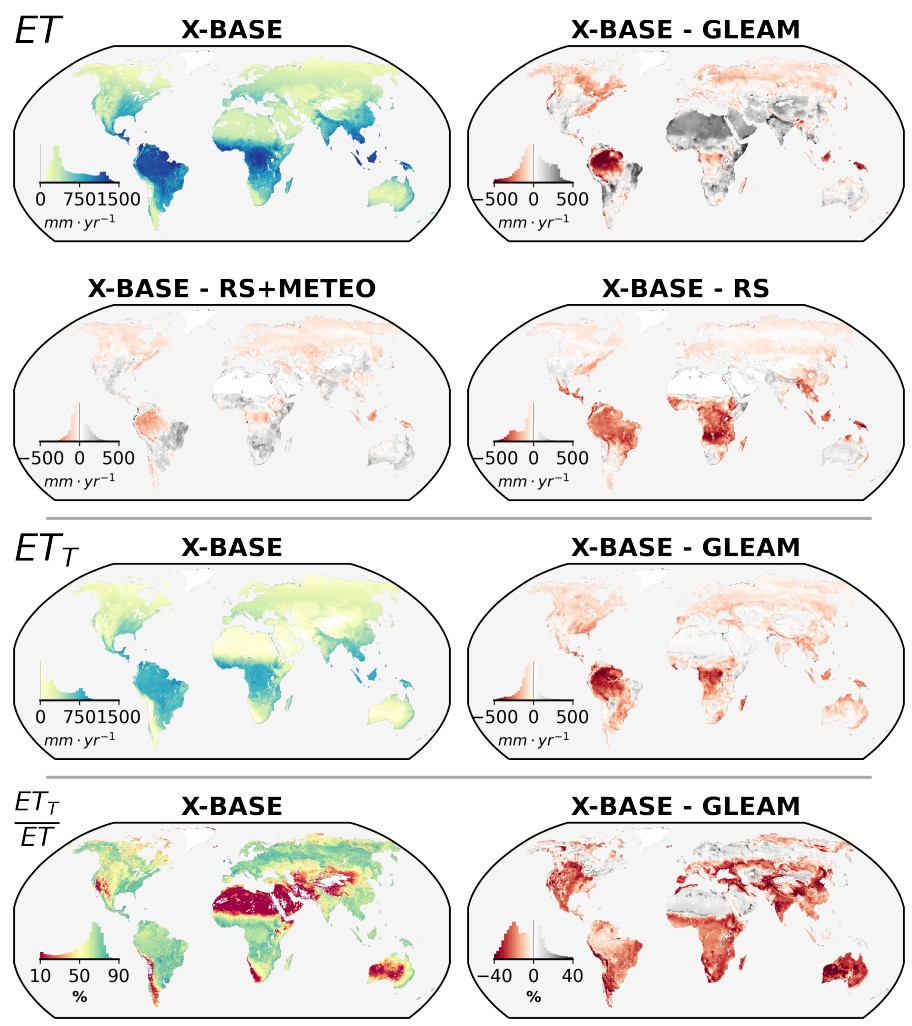

**Figure 8. Comparison of evaporative flux estimates** $ET$, $ET_T$ **and** $ET_T/ET$ **from X-BASE and its difference with RS, RS+METEO, and GLEAM.** $ET_T$ is compared in the case of GLEAM, but is unavailable in the previous FLUXCOM ensembles.



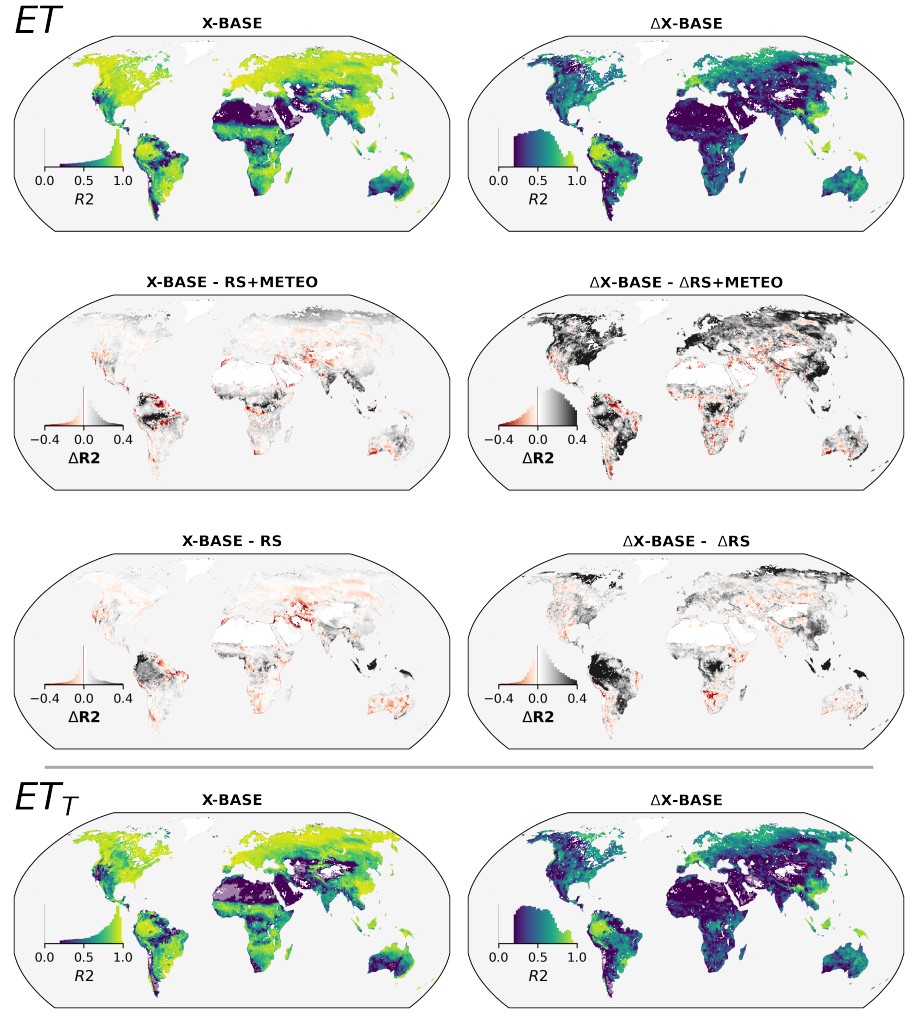

**Figure 9.** $R^2$ (computed as the square of the Pearson correlation) between X-BASE $ET$ and GLEAM $ET$ for the actual time series (left column) and anomalies from the median seasonality (right column). The middle panels compare the agreement between X-BASE $ET$ and GLEAM $ET$ to the agreement between FLUXCOM $ET$ and GLEAM $ET$. The bottom panel shows the squared correlations between X-BASE and GLEAM $ET_T$, but no comparisons to FLUXCOM because FLUXCOM did not include $ET_T$. All comparisons are done for time series with a resolution of 16 days and 0.05 degrees for the years 2001-2020. Semi-transparent areas mark pixels in which the correlation of at least one of the data sets is negative.





## 4 Discussion

### 4.1 Higher consistency of $NEE$ with atmospheric carbon cycle constraints

Although FLUXCOM-X follows the same fundamental approach as FLUXCOM, we find a substantial improvement of the magnitude of the annually integrated $NEE$ of FLUXCOM-X-BASE over previous FLUXCOM products (Jung et al., 2020)
when compared to independent estimates from atmospheric inversions. The mean global X-BASE $NEE$ of -5.75 $PgC \cdot yr^{-1}$ is slightly smaller than the inferred $NEE$ of -3.92 $PgC \cdot yr^{-1}$ (corrected for fire emissions based on GFED 4.1) from CarboScope. The remaining difference could easily be explained by carbon sources such as aquatic evasion and volatile organic compounds that are included in the atmospherically based estimate but not in eddy-covariance based FLUXCOM (see Jung et al. (2020) and Zscheischler et al. (2017) for further discussion).

The improved global $NEE$ of FLUXCOM-X-BASE originates most likely from enhanced quality of eddy covariance measurements in the training. Previous up-scaling-based $NEE$ products of (Jung et al., 2011, 2020; Bodesheim et al., 2018) - all based on the La Thuile FLUXNET dataset but varying with respect to machine learning methods, predictor variables, and temporal resolution - consistently estimated a nearly three-fold larger global terrestrial carbon uptake compared to X-BASE. As discussed and speculated in Jung et al. (2020), La Thuile likely contained biased $NEE$ measurements, in particular for
some tropical sites (Fu et al., 2018), and together with the sparsity of data in the tropics, these biases were propagated to unrealistic tropical and global $NEE$ estimates. The fact that we can now reconcile bottom-up global eddy-covariance-based $NEE$ and estimates from top-down atmospheric inversions is a major achievement of the FLUXNET community. For context: 1 $PgC \cdot yr^{-1}$ over the global vegetated area ($145 \times 10^6 km^2$) corresponds to ~7 $gC \cdot m^2 \cdot yr^{-1}$, which marks a challenge for achieving such accuracy of mean $NEE$ at any one flux tower site. The lesson learned here emphasizes once more that it is
crucial to control for and minimize systematic biases of in-situ eddy covariance measurements (Moncrieff et al., 1996).

The improved seasonality of X-BASE $NEE$, in particular for boreal regions, likely also results from enhanced information in the training data due to the hourly resolution. Similar improvements were observed in Bodesheim et al. (2018) who extended RS+METEO by training on half-hourly flux observations. The hourly resolution improves the seasonal high-latitude $NEE$ likely due to better capturing the responses to light when daylength varies strongly.

### 390 4.2 New opportunities by X-BASE products

The improvements of $NEE$ make X-BASE attractive as a data-driven biogenic prior for atmospheric inversions (Munassar et al., 2022). Moreover, its hourly resolution facilitates better integration in inversion systems due to the accounting of diurnal flux and atmospheric transport variations, while its high spatial variations can provide patterns of flux variations that cannot be resolved by atmospheric constraints alone.
For the first time, X-BASE includes a global data-driven product of ecosystem transpiration. The estimated global $ET_T/ET$ ratio of 57% is consistent with independent top-down assessments from isotope base methods (Good et al., 2015; Coenders-Gerrits et al., 2014) and past up-scaling estimates (Wei et al., 2017; Schlesinger and Jasechko, 2014). The spatially and temporally high resolution data-driven X-BASE $ET_T$ product provides a valuable complementary perspective to simulations from





process-based land surface models, which show large disagreements and often indicate global $ET_T/ET$ below 50% (Berg

and Sheffield, 2019; Miralles et al., 2016). This advancement opens new opportunities for large scale studies of carbon- water
relations on a diurnal time scale. The generation of the X-BASE $ET_T$ product was facilitated by the development of site-level
evapotranspiration partitioning methods (Nelson et al., 2018, 2020) underlining once more the importance of advances by the
FLUXNET community for Earth system science.

## 5 Tackling persistent challenges

Next to these improvements and opportunities, we find that some key issues previously identified in FLUXCOM (Tramontana
et al., 2016; Jung et al., 2020; Bodesheim et al., 2018) persist in X-BASE. These include the insufficient representation of
water-related effects, the limited predictability of the spatial patterns of mean $NEE$, as well as severe limitations with respect
to the variability between years and over decades. The overestimation of mean $ET$ in very dry, sparsely vegetated areas (Fig.
B3), as well as the poorer consistency of $NEE$ seasonality with inversions in water limited regions (Fig. 5) illustrate the

persistent challenge and importance of capturing water effects on land-atmosphere flux variations. For $GPP$ temporal patterns
we find that X-BASE shows improved agreement with SIF in water limited regions compared to RS+METEO (Fig. 7), which
is likely because X-BASE uses concomitantly changing remote sensing observations opposed to a mean seasonal cycle only
in RS+METEO. However, X-BASE shows deteriorated agreement with SIF when compared to RS, even though X-BASE was
trained on hourly flux observations with improved coverage of dry conditions (Fig. 1). This decrease in performance indicates

clearly the importance and uncertainty related to the predictor variable set for capturing water related effects. Thus, there is
considerable potential for advancements by including remote sensing based predictors on soil moisture, sub-daily varying land
surface temperature from geostationary satellites, SIF, and vegetation optical depth. Here, a key challenge resides in achieving
a sensible integration of flux observations with footprints that are much smaller than corresponding Earth observation products.

Missing predictor variables is likely also a main reason for the limited skill of predicting between site variability of mean

$NEE$ (Fig. 2), which can depend on legacy effects of disturbances and management that are not accounted for. Novel and
complementary Earth observation products that characterise ecosystem structure and states related to biomass and canopy
heights from SAR and LIDAR should help raise the accuracy of FLUXCOM-X based mean $NEE$ in future efforts. X-BASE
shows a prominent pattern of carbon flux to the atmosphere in sub-tropical and crop dominated regions of India and the Sahel,
which emphasizes the need to improve in-situ data coverage for agricultural systems, especially outside the temperate zone,

and including important meta-data to better characterize these ecosystems and their site history. Despite the greatly reduced
overall bias of mean $NEE$, we emphasize that X-BASE products are pre-mature for diagnosing spatial variations of mean
$NEE$.

The representation of longer term dynamics remains an area with opportunity for improvements in X-BASE. Inter-annual
variability is still poorly reproduced in cross validation (Fig. 2), particularly for $NEE$, which is likely not only due to the

complexity of processes shaping inter-annual variations but also due to temporal discontinuities in flux tower time series related
to changes in instrumentation and factors like management (Jung et al., 2023) that are not accounted for. The complexities of



relying on field deployed instrumentation, together with the uncertainties related to linking satellite and flux data, causes poor signal to noise ratios and may impede good cross-validation results for i.a.v.. Globally, comparisons of X-BASE with inversions reveal an underestimated inter-annual variance and a poor correlation for global $NEE$ i.a.v. (Fig. 5 and Table B2). Interestingly, X-BASE $GPP$ shows improved correspondence with SIF anomalies compared to RS+METEO, especially in water limited regions (Fig. 7), while no such improvement is evident for global $NEE$, which is likely due to the compensatory water effects in the global $NEE$ signal (Jung et al., 2017). That a comparison of RS+METEO runs with different meteorological forcing data showed the weakest correspondence with inversion i.a.v. when using ERA5 (Jung et al., 2020) explains the substantially better correlations of RS+METEO with inversions for $NEE$ i.a.v. in earlier studies (Jung et al., 2017, 2020), and may also explain the poor correlation of X-BASE. Thus, testing if alternative meteorological forcing data can improve global $NEE$ i.a.v. for X-BASE is an important next step. It remains unclear at this point if accurate inter-annual variations at site-level and globally can be achieved by the FLUXCOM approach in the near future. Additional constraints beyond FLUXNET such as atmospheric $CO_2$ measurements (Upton et al., 2023) or theoretical considerations in the form of hybrid (Reichstein et al., 2019) or deep learning models (Camps-Valls et al., 2021) are promising and such endeavours should be fostered.

## 6 Conclusions

We presented X-BASE, a new set of global high-resolution data-driven products of land-atmosphere fluxes from the FLUX-COM approach. This represents a cornerstone of our developments of the FLUXCOM-X framework designed to explore and mitigate current limitations to up-scaling from site to global scale. Improvements of the eddy covariance data facilitated reconciling estimates of global terrestrial net carbon exchange from X-BASE with top-down atmospheric inversions, and allowed for the first time the generation of a global data-driven estimate of ecosystem transpiration. Beyond fostering all activities to enhance quality and coverage of available flux tower observations, most promise for future advancements by FLUXCOM-X relates to the synergistic exploitation of complementary satellite data streams to better capture water-related, site-history, and management effects. This will be challenging as it requires developing strategies and methodologies to better integrate in-situ flux observations and spaceborne Earth observations with very heterogeneous acquisition properties and with spatial resolutions that are often very coarse compared to flux tower footprints. The recent de-orbiting of the TERRA spacecraft requires employing alternative satellite missions where practical issues of data acquisition and conceptual issues related to temporal consistency and reduced overlap with FLUXNET records pose imminent challenges. With FLUXCOM-X we have prepared the ground for tackling these challenges which can facilitate up-to-date and accurate flux estimates and thereby contribute to increased understanding of the Earth system in the future.





## Appendix A: Details on processing of Earth Observation Data

### A1 Dynamic quality control and cutout size

The conditions in the pixels around a given EC station should best represent the conditions of the land surface in the area where the actual fluxes originate from. Given that the actual flux footprints are not generally available or computable for lack of critical information, we assume that the pixel containing the actual EC station (the 'tower pixel') is most representative for the dynamics of the area of influence on a tower. However, data availability and quality in the tower pixel is often insufficient. An iterative approach therefore selects both the cutout size and the strictness of the BRDF inversion quality from within defined bounds in a way that maximizes data availability and that ensures representativeness of the spatially averaged time series for the given site at the same time. In more detail, we start with a strict criterion for BRDF inversion quality (BRDF_Albedo_Band_Quality_Bandx flag in MCD43A2 <= 2, meaning only full inversions). Then three options regarding the cutout size are considered:

A) only the tower pixel,

B) those 20% of pixels within 4x4 km$^2$ around a tower that are best correlated with the tower pixel are linearly regressed against the tower pixel and subsequently spatially averaged,

C) the 25% of pixels within a 4x4 km$^2$ area that are closest to the tower are averaged with the inverse of the distance to the tower as weight.

The criteria for selection between options A-C is based on the number of available good quality observations n in the resulting spatial average time series per site as follows:

```
if (n_A >= 60 %) & (n_B <= 70 %):
    select A
elif (n_A >= 60 %) & (n_B >= 70 %):
    select B
elif (n_A < 60 %) & (n_A > 15 %):
    select B
else:
    select C
```

If after the previous steps still less than 40% of good quality observations outside of snow covered times are available in the resulting average time series for a given site and index, the BRDF inversion quality threshold is relaxed to also allow magnitude inversions (MCD43A2 BRDF inversion quality flag <= 3), and the procedure to select the pixels contributing to the average described above is repeated. Consequently, the size of the area that a MODIS reflectance time series represents varies between sites, and so does the BRDF inversion quality.





For the global gridded MODIS data, the BRDF inversion quality is consistently selected as <=2 or <=3 based on the number of available good quality observations in a pixel.

## A2   Details on the treatment of land cover information

Land cover information was passed through an intermediary classification system to both act as an encoding mechanism and to allow for arbitrary links between classification schemes. Rather than simple true/false classification for each category, different attributes are classified based on whether the classification has (value=1.0), might have (value=0.5), does not have (value=0.0) a specific feature, or is unknown (value=-1.0). In the specific case of the MCD12Q1 classificaiton scheme, the conversion is as seen in Table A1.

Table A1: **Land cover intermediary classification encoding for MCD12Q1 classifications.**

|  | Trees | Shrubs | Grasses | Crops | Unveg | Water | Wetland | C4_photo | Managed | Needleleaf | Broadleaf | Deciduous | Evergreen |
|---|---|---|---|---|---|---|---|---|---|---|---|---|---|
| ENF | 1 | 0 | 0 | 0 | 0 | 0 | 0 | 0 | -1 | 1 | 0 | 0 | 1 |
| EBF | 1 | 0 | 0 | 0 | 0 | 0 | 0 | 0 | -1 | 0 | 1 | 0 | 1 |
| DNF | 1 | 0 | 0 | 0 | 0 | 0 | 0 | 0 | -1 | 1 | 0 | 1 | 0 |
| DBF | 1 | 0 | 0 | 0 | 0 | 0 | 0 | 0 | -1 | 0 | 1 | 1 | 0 |
| MF | 1 | 0 | 0 | 0 | 0 | 0 | 0 | 0 | -1 | -1 | -1 | -1 | -1 |
| CSH | 0.5 | 1 | 0 | 0 | 0 | 0 | 0 | -1 | -1 | -1 | -1 | -1 | -1 |
| OSH | 0 | 1 | 0.5 | 0 | 0.5 | 0 | 0 | -1 | -1 | -1 | -1 | -1 | -1 |
| WSA | 1 | 0.5 | 0.5 | 0 | 0 | 0 | 0 | -1 | -1 | -1 | -1 | -1 | -1 |
| SAV | 0.5 | 0.5 | 1 | 0 | 0 | 0 | 0 | -1 | -1 | -1 | -1 | -1 | -1 |
| GRA | 0 | 0 | 1 | 0 | 0 | 0 | 0 | -1 | -1 | 0 | 0 | 0 | 0 |
| SNO | -1 | -1 | -1 | -1 | 0 | 0.5 | 1 | 0 | 0 | -1 | -1 | -1 | -1 |
| CRO | 0 | 0 | 0 | 1 | 0 | 0 | 0 | -1 | 1 | 0 | 0 | 0 | 0 |
| WET | 0 | 0 | 0 | 0 | 0 | 1 | 0 | 0 | 0 | 0 | 0 | 0 | 0 |



# Appendix B: Additional results

## B1  Additional cross-validation results

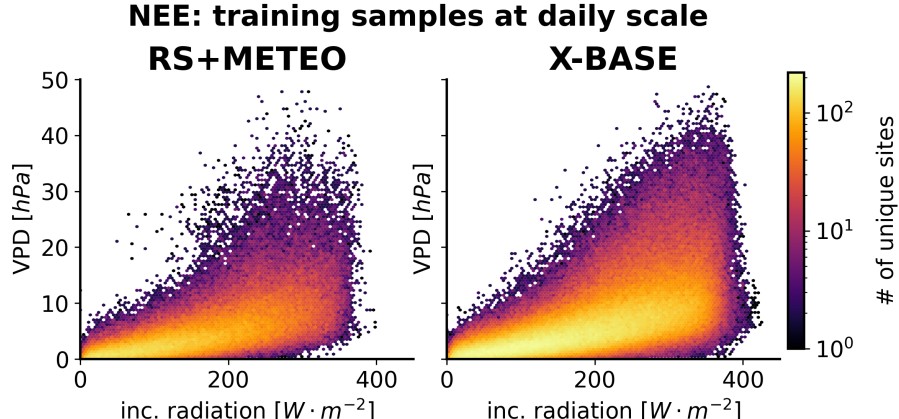

**Figure B1. Cross-validation sampling in meteorological space:** Number of unique sites contributing to sampling for $NEE$ for FLUXCOM RS+METEO (left) compared to the sampling in the X-BASE set-up (right). Color corresponds to number of unique sites per bin in log scale.





## B2   Large carbon uptake in tropical croplands

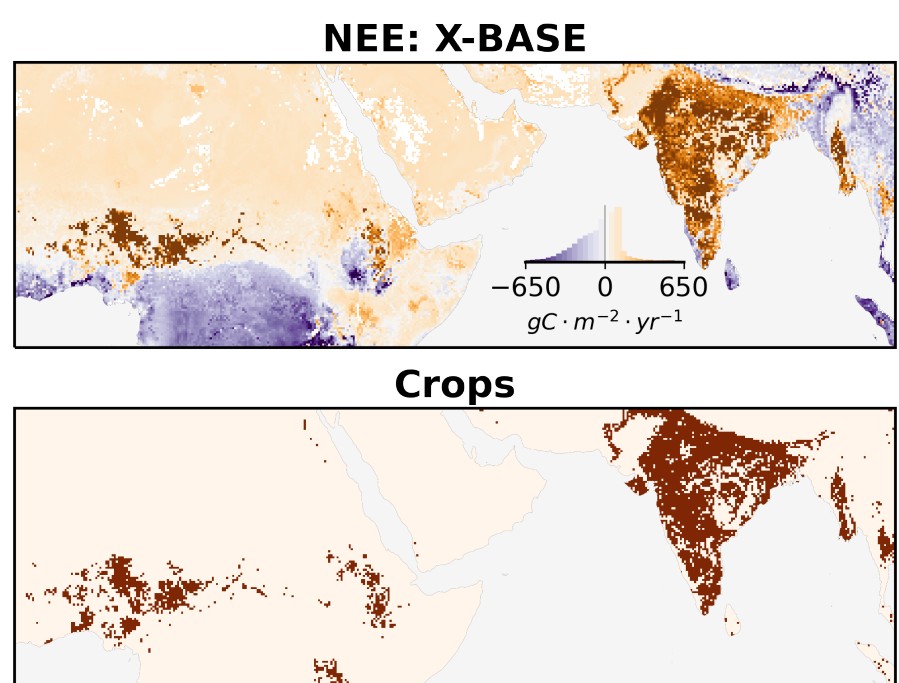

Figure B2. Large carbon uptake in tropical croplands



## B3 Global magnitude of all fluxes

Table B1: **Global magnitude of all fluxes.** Column *Global Total* is the globally integrated flux for all areas including sparsely vegetated dry areas from 2001-2020. The column *Veg. Areas* includes a common mask which removes sparsely vegetated areas which are not computed for the RS and RS+METEO products. Values reported after the $\pm$ correspond to the standard deviation across years.

|  | Global Total | Veg. Areas |
|---|---|---|
| **NEE** | | |
|  | $PgC \cdot yr^{-1}$ | $PgC \cdot yr^{-1}$ |
| X-BASE | $-5.75 \pm 0.33$ | $-7.12 \pm 0.32$ |
| RS+METEO | - | $-21.27 \pm 0.59$ |
| RS | - | $-19.08 \pm 0.93$ |
| CarboScope | $-3.88 \pm 0.84$ | $-3.92 \pm 0.84$ |
| **GPP** | | |
|  | $PgC \cdot yr^{-1}$ | $PgC \cdot yr^{-1}$ |
| X-BASE | $124.7 \pm 2.1$ | $121.9 \pm 2.0$ |
| RS+METEO | - | $121.6 \pm 0.4$ |
| RS | - | $113.2 \pm 1.8$ |
| **ET** | | |
|  | $km^3 \cdot yr^{-1}$ | $km^3 \cdot yr^{-1}$ |
| X-BASE | $74.7\text{x}10^3 \pm 0.9\text{x}10^3$ | $68.9\text{x}10^3 \pm 0.9\text{x}10^3$ |
| RS+METEO | - | $68.3\text{x}10^3 \pm 0.3\text{x}10^3$ |
| RS | - | $78.5\text{x}10^3 \pm 0.5\text{x}10^3$ |
| GLEAM | $72.5\text{x}10^3 \pm 1.0\text{x}10^3$ | $70.9\text{x}10^3 \pm 0.9\text{x}10^3$ |
| **ET$_T$** | | |
|  | $km^3 \cdot yr^{-1}$ | $km^3 \cdot yr^{-1}$ |
| X-BASE | $42.6\text{x}10^3 \pm 1.0\text{x}10^3$ | $41.8\text{x}10^3 \pm 0.9\text{x}10^3$ |
| GLEAM | $50.7\text{x}10^3 \pm 0.6\text{x}10^3$ | $50.7\text{x}10^3 \pm 0.6\text{x}10^3$ |
| **ET$_T$/ET** | | |
| X-BASE | $57.0\% \pm 0.6\%$ | $60.7\% \pm 0.6\%$ |
| GLEAM | $70.0\% \pm 0.6\%$ | $71.4\% \pm 0.6\%$ |



## B4 Linear trends and inter-annual variability for all fluxes

Table B2: **Long-term variability of fluxes.** Column *linear trend* is the linear slope of annually integrated fluxes over the years 2001-2020. The column *inter-annual variability* is computed as the standard deviation of annually integrated fluxes after the trend is removed.

|  | Linear Trend | Inter-annual Variability |
|---|---|---|
| **NEE** | | |
| | $PgC \cdot yr^{-2}$ | $PgC \cdot yr^{-1}$ |
| X-BASE | 0.017 | 0.306 |
| RS+METEO | 0.095 | 0.229 |
| RS | -0.129 | 0.557 |
| CarboScope | 0.006 | 0.837 |
| **GPP** | | |
| | $PgC \cdot yr^{-2}$ | $PgC \cdot yr^{-1}$ |
| X-BASE | 0.340 | 0.575 |
| RS+METEO | -0.053 | 0.246 |
| RS | 0.248 | 1.023 |
| **ET** | | |
| | $km^3 \cdot yr^{-2}$ | $km^3 \cdot yr^{-1}$ |
| X-BASE | $0.144 \times 10^3$ | $0.331 \times 10^3$ |
| RS+METEO | $-0.010 \times 10^3$ | $0.301 \times 10^3$ |
| RS | $0.053 \times 10^3$ | $0.392 \times 10^3$ |
| GLEAM | $0.102 \times 10^3$ | $0.730 \times 10^3$ |
| **$ET_T$** | | |
| | $km^3 \cdot yr^{-2}$ | $km^3 \cdot yr^{-1}$ |
| X-BASE | $0.158 \times 10^3$ | $0.277 \times 10^3$ |
| GLEAM | $0.035 \times 10^3$ | $0.596 \times 10^3$ |
| **$ET_T/ET$** | | |
| | $\% \cdot yr^{-1}$ | $\%$ |
| X-BASE | 0.102% | 0.157% |
| GLEAM | -0.054% | 0.452% |



## B5    Potential overestimation of $ET$ in dryland areas

Maps in Fig. B3 show the extend where $ET$ and $ET_T$ exceed precipitation as the ratio between the total of each flux to total precipitation (from GPCC Schneider et al. (2022)). Overall, X-BASE $ET$ largely exceeds precipitation in most dry, sparsely vegetated areas, indicating overestimation. In contrast, $ET_T$ does not show such extensive overestimation, limited instead to only smaller regions of the Sahara.

The amount of overestimation of X-BASE $ET$ can be roughly estimated by replacing areas where annual $ET$ exceeds pre-
cipitation inputs with the corresponding annual precipitation inputs for each grid cell, i.e. replacing areas where the $ET/precip$ ratio is more than a threshold with the precipitation rather than the estimated $ET$. Using thresholds from 1.25 to 2.5 gives an excess of $ET$ (i.e. original $ET$ minus precipitation corrected) from 3.9x10$^3$ to 6.1x10$^3$ $km^3 \cdot yr^{-1}$.



# X-BASE ET ratio to Precip.

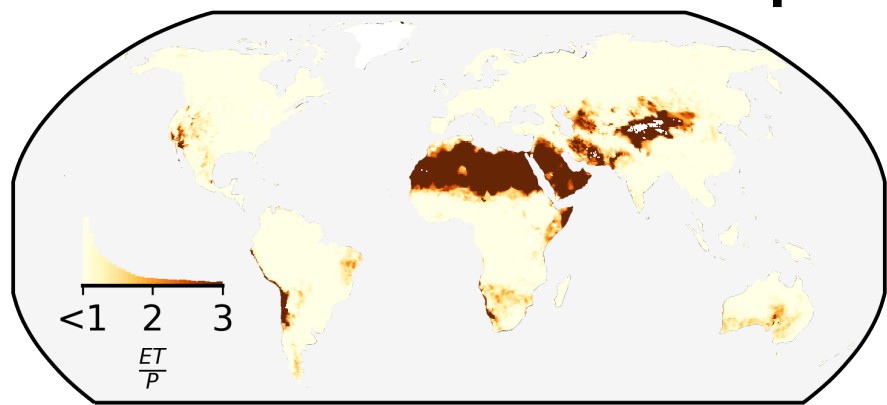

# X-BASE ET$_T$ ratio to Precip.

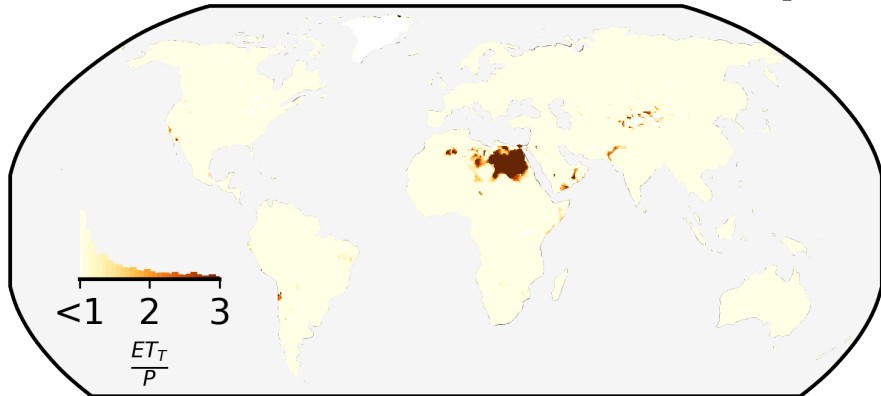

**Figure B3. Potential** $ET$ **overestimation** based on the ratio of estimated $ET$ to precipitation from the Global Precipitation Climatology Centre (GPCC Schneider et al. (2022)).



*Data availability.* All data is available as aggregated NetCDF file formats, to ease data handling for common use cases, from the ICOS Carbon Portal (https://doi.org/10.18160/5NZG-JMJE). Furthermore, the full resolution data is accessible in the zarr format and in a publicly
available object store provided by German Climate Computing Center (Deutsches Klimarechenzentrum, DKRZ). Instructions on how to all data, as well as the full dataset, can be found at the associated repository (https://gitlab.gwdg.de/fluxcom/fluxcomxdata).

*Author contributions.* JN, SW and MJ wrote the manuscript, and all authors contributed to the editing and structuring of it. JN, SW, FG, BK, ZH wrote the code which constitutes the modeling environment FLUXCOM-X, structuring it according to discussions including MJ, UW, GD, and MM. Editorial input was provided by AI, AK, AK, AS, BB, BK, DB, DP, FT, GCV, GD, GN, GW, JA, JC, KI, KMK, KN, LH, LS,
MK, MM, MR, NB, RLS, SS, TT, and WZ. All other authors contributed through their work at one or more eddy-covariance stations through the acquisition of funding, set-up and maintenance, pre-processing and quality control, administration, and their insertion into regional or global networks.

*Competing interests.* We declare that one or more of the coauthors of this manuscript is a member of the editorial board of Biogeosciences.

*Acknowledgements.* We would like to thank the broader eddy covariance community, including FLUXNET and the associated regional
networks, particularly the European Integrated Carbon Observation System (ICOS) and AmeriFlux. We also acknowledge the contributions of Andrej Varlagin and colleagues to these efforts. We thank the team at the ICOS Carbon Portal for their support in publishing the FLUXCOM-X data sets, with great thanks in particular to Ute Karstens and Zois Zogopoulos. We also thank Brendan Byrne and colleagues of Jet Propulsion Laboratory, California Institute of Technology, Pasadena, CA, USA for use of the OCO2 data. The work of S. Walther, J. A. Nelson and M. Jung was funded in part by European Union's Horizon 2020 research and innovation program (grant no. 776186 CHE,
776810 VERIFY, 958927 CoCO2,820852 E-SHAPE). S. Walther acknowledges funding from an European Space Agency Living Planet Fellowship in the project Vad3e mecum as well as the CCI LST project (4000123553/18/I-NB). G. Duveiller and Z. M. Hamdi acknowledge funding from the European Space Agency in the Sen4GPP project (4000134598/21/I-NB). G. Duveiller acknowledges Horizon Europe funding (Open Earth Monitor Cyberinfrastructure project, 101059548). D. Zona acknowledges NSF award numbers 2149988 and 1932900. A. Klosterhalfen and A. Knohl acknowledge funding by the German Federal Ministry of Education and Research (BMBF) as part of the
European Integrated Carbon Observation System (ICOS), by the Deutsche Forschungsgemeinschaft (INST 186/1118-1 FUGG) and by the Ministry of Lower-Saxony for Science and Culture (DigitalForst: Niedersächsisches Vorab (ZN 3679)). L. Montagnani acknowledges funding provided by Forest Services, Autonomous Province of Bolzano. E. Yepez acknowledges that MX-Tes is part of the MexFlux regional network. Funding for the Swiss sites is greatly acknowledged from various sources: from the EU project SUPER-G (contract no. 774124), the SNF projects M4P (40FA40_154245), DiRad (146373), InnoFarm (407340_172433), CoCo (200021_197357), ICOS-CH (20FI21_148992,
20FI20_173691, 20FI20_198227) and InsuranceGrass (100018L_200918), from NESTLE via the ETH foundation (DONA), from the ETH Board and from ETH Zurich (project FEVER ETH-27 19-1). Funding for US-BZB, US-BZF, US-BZo, and US-BZB was provided by National Science Foundation Grants DEB LTREB 1354370 and 2011257, DEB-0425328, DEB-0724514, and DEB-0830997, as well as funding by the US Geological Survey Climate R&D program. Bonanza Creek Long Term Experimental Research station provided lab space, equip-



ment. US-ICs and US-ICt were supported by the grants from the Arctic Observatory Program of the National Science Foundation (grant
numbers 1936752, 1503912, 1107892). S. Aranda-Barranco acknowledges the projects PID2020-117825GB-C21 and PID2020-117825GB-
C22 funded by MCIN/AEI/10.13039/501100011033, as well as support by the FPU grant by the Ministry of Universities of Spain (REF:
FPU19/01647). SE-Deg, SE-Svb and SE-Ros acknowledge funding from the Swedish Research Council, and contributing research insti-
tutes to the Swedish Integrated Carbon Observation System (ICOS-Sweden) Research Infrastructure and the Swedish Infrastructure for
Ecosystem Science (SITES). T. Tagesson was funded by the Swedish National Space Agency (SNSA Dnr 2021-00144), and FORMAS
(Dnr. 2021-00644). W. Woodgate is supported by an Australian Research Council DECRA Fellowship (DE190101182). I. Mammarella
acknowledges funding from Academy of Finland (N-PERM 341349), ICOS-Finland UH, EU projects (GreenFeedback 101056921, Liwe-
For). D. Vitale acknowledges the Integrated Carbon Observation System - Research Infrastructure (ICOS ERIC, https://www.icos-cp.eu/)
and the ICOS ETC funding from the Italian Ministry of Research. M. Göckede was supported by the European Research Council (ERC)
under the European Union's Horizon 2020 research and innovation programme (grant agreement No 951288, Q-Arctic). S. Sabbatini ac-
knowledges OEMC project - Grant agreement ID: 101059548. The DE-Geb site received funds within the ICOS Germany preparatory and
implementation phase by the Federal Ministry of Education and Research and is supported by the Ministry of Digital and Traffic through
ICOS station contributions as well as by the Ministry of Food and Agriculture covering operational costs. G. Gerosa thanks the Catholic
University of Brescia for continuous supporting the research station of Bosco Fontana (ICOS station IT-Bft). L. Šigut acknowledges support
by the Ministry of Education, Youth and Sports of CR within the CzeCOS program (grant number LM2023048). G. Camps-Valls would like
to acknowledge the support from the European Research Council (ERC) under the ERC Synergy Grant USMILE (grant agreement 855187).
A. Desai acknowledges the US Dept of Energy American Network Management Project award to ChEAS core site cluster (US-PFa, US-
WCr, US-Syv, US-Los), Wisconsin Potato and Vegetable Growers Association and WI Dept of Natural Resources (US-CS*), NSF 1822420,
2313772 (US-PFa). M. Roland and B. Gielen acknowledge the Research Foundation Flanders (FWO) for the support of ICOS research in-
frastructure in Flanders, Belgium. D. Papale thanks the support of the ITINERIS - Italian Integrated Environmental Research Infrastructures
System project (IR0000032) funded by Next Generation EU Mission 4.2.3.1. T. Vesala acknowledges ICOS-Finland (University of Helsinki)
and Flagship funding (grant no. 337549). T. Sachs acknowledges that the DE-Zrk site relies on infrastructure of the Terrestrial Environmental
Observatories Network (TERENO) supported by a Helmholtz Young Investigators Grant (VH-NG-821). E. Pendall acknowledges Australian
Terrestrial Ecosystem Research Network, as part of the National Cooperative Research Infrastructure System. S. Knox was also support by
an NSERC Discovery Grant (RGPIN-2019-04199) and Alliance Grant (ALLRP 555468-20). H. Kobayashi acknowledges ArCSII No. JP-
MXD1420318865 (US-Prr). S. Metzger acknowledges the National Ecological Observatory Network, which is a program sponsored by the
National Science Foundation and operated under cooperative agreement by Battelle. This material is based in part upon work supported
by the National Science Foundation through the NEON Program. B. Heinesch and C. Vincke acknowledge the Service Public de Wallonie
(SPW-DGO6) for the support of ICOS research infrastructure in Wallonia, Belgium.



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
