# Peer review of "X-BASE: the first terrestrial carbon and water flux products from an extended data-driven scaling framework, FLUXCOM-X"

_EGUsphere, 2024_

## Author Response (AR1)

**Response to reviewer 1**

**Overview**

To upscale eddy fluxes, specifically water and carbon, globally, the authors used a data-driven modeling framework based on FLUXCOM, and produced a new global product, X-BASE. The paper is well written, however, there are few aspects that need to be addressed. First, I understand and agree that the product is the initial basic version, but I am not seeing the highlight of it compared to FLUXCOM. Second, X-BASE product was compared to many other products, but it would be nice to have a defined truth dataset for comparisons, otherwise, it does not really showcase the advantage of X-BASE products. Last, since the model relies on field measurements (i.e., eddy covariance), the introduction and discussion would benefit from further reviews. Please see my comments below for detailed explanations. Overall, thank you for your dedication and hard work to advance the data needs for the flux community, and I look forward to reviewing your revision and using your products.

*We thank you very much for your in depth review and encouraging words. We wrote the paper with the intention to give an outlook on the products which is both technically complete and understandable to a wide set of potential users, so the points that you raised indeed provide a good reflection on the key aspects to help clarify and strengthen the paper.*

*Much of the work is indeed rather technical, building an improved up-scaling structure with the technical capabilities to go from hundreds of independent field measurements spanning decades to a globally integrated product. The new framework was built not just to provide a suite of new products, but to have the ability to experiment and improve on a time horizon of months rather than years. Thus our hope is that major advancements will be forthcoming using the technology, and thus the results here are more of a benchmark. Still, compared to FLUXCOM the X-BASE products represent major advancements which are 1) the reduction of the estimated global carbon uptake (which is now in considerably better alignment with estimates from atmospheric inversion products), 2) the high spatial AND high temporal resolution, and 3) the inclusion of a new variable, transpiration.*

*Second, X-BASE is one of several approaches to estimate terrestrial fluxes, all complementary to each other and with their individual strengths and weaknesses. A reference truth dataset does not exist and we refrain from defining one, but rather test for robust variability patterns by comparing to estimates from other independent models. Lastly, we agree on the importance of eddy covariance measurements and on giving more space for relevant information and discussion points in the paper to them, and have done so in a revised paper version.*

*We address your concerns in more detail in the comments below.*

**Main comments:**

1. What do you mean by EC as near continuous (L26)? Aside from data gaps, isn't EC continuously measuring the flux exchanges? Additionally, please clarify that the flux variations (L27) are built on high frequency sampling (10Hz, 20HZ, etc). Then the half hourly data comes in to form that long-term data record.

*Yes indeed, we only wish to highlight the fact that while the measurements can be made continuously, issues such as system malfunction and non-turbulent conditions result in gaps in the data. We have made additional clarifications (see response to #2 below).*

2. In the paragraph starting from L68, you started introducing the use of machine learning, which relies on training data. If these data are ingested into ML models with uncertainty, how would you justify this? There are many existing issues for eddy covariance, ranging from theoretical to practical challenges, but EC data are widely used. Consider adding a short review on EC data quality, and you can orbit the short review here around the closure issue given you are using FLUXNET2015. Here are some papers to consider:

- https://www.sciencedirect.com/science/article/pii/S016819231100339X?via%3Dihub
- Advection:
    - EBEX: https://link.springer.com/article/10.1007/s10546-007-9161-1
    - Kyaw Tha Paw U's group: https://www.sciencedirect.com/science/article/pii/S0168192311000220
    - Dennis Baldocchi's group: https://papers.ssrn.com/sol3/papers.cfm?abstract_id=4587556
- There are few other issues but can be found in these papers.

*Yes, data biases and uncertainties, including energy balance non-closure and advection, are a key issue which influences site level budget and potentially also the resulting global products. As the up-scaling exercise incorporates many different aspects (eddy covariance and meteorological measurements, remote sensing, machine learning, model evaluations, etc.) we tried to give a relevant overview of all key components without overwhelming the reader. But as the reviewer has pointed out, eddy covariance is a foundation of the methodology and warrants more information on the related uncertainties early in the manuscript. To that end we have added a short paragraph to the introduction. However regarding the carbon fluxes and the advection, the robust n-folds validation done in Tramontana et al. 2016 shows that the ML system is quite robust to the uncertainty.*

3. Similar to my first point in the overview, X-BASE certainly has advantage with enhanced spatial and temporal coverage and better training data (L87)

- What is the main innovation of X-BASE? Can you highlight the advantages compared to FLUXCOM and other upscaling concepts?

- I may not be understanding it right, but X-BASE is the same as FLUXCOM-X, except X-BASE is the initial result. I keep getting confused between X-BASE and FLUXCOM-X when I was reading the paper, consider adding a table to describe FLUXCOM, FLUXCOM-X, X-BASE, including spatial and temporal resolutions.

*We have tried to come up with naming conventions for frameworks and data products and to associate them with their key innovations, but failed to clearly communicate them. So thank you for your questions in this respect.*

*FLUXCOM-X is what we call the up-scaling environment that we have implemented, and which - in contrast to FLUXCOM and other up-scaling implementations - has the key innovation of being fully flexible and scalable. This allows regular updates of EC data both in space and time, training and predicting at different resolutions, ingesting additional/ removing any predictor data set (thus exploiting different space-based missions for example), using different machine learning methods. Any data product with flux estimates that will be generated and published from the modelling environment FLUXCOM-X will follow the naming convention X-[specific name]. X-BASE is as you write the first, very basic product produced from FLUXCOM-X. Compared to the set-up used in FLUXCOM, the key innovations are:*

- *high spatial and high temporal resolution*
- *meteorological and satellite-based predictors both resolved in time*

- *training data base is larger and has higher quality*
- *a new product: transpiration*

*These innovations already allow addressing new scientific questions (such as regarding diurnal dynamics or trends related to meteorology or vegetation itself) and improve the flux products (as indicated by the reduced global NEE). Otherwise X-BASE is very similar to the set-up in FLUXCOM, and this is intentional, as we are interested in the effect that these basic changes already have on the flux estimates, in order to then guide the focus of future development with FLUXCOM-X.*

*We have rewritten the last two paragraphs of the introduction in order to clarify on the naming conventions and to more clearly point out the differences and similarities between FLUXCOM products and X-BASE products. We have also added an overview in the supplement in case there is still confusion.*

4. Thank you for not forcing the closure! For the Transpiration Estimation Algorithm (L116), please consider spilling more information. What is the basic pipeline? What are the assumptions? The additional information could help readers understand better.

*Yes, indeed we find that energy balance closure is a very important issue in this work and warrants a more focused study, which the new framework (FLUXCOM-X) will be able to facilitate. Regarding the Transpiration Estimation Algorithm (TEA), indeed the methodology was not well documented in this version. We have added a brief outline of how the methodology works here where it is introduced, which also points readers to more information. In brief, TEA uses the relationship between GPP and ET from periods where surface evaporation is likely to be minimized to isolate the expected water use efficiency as GPP to transpiration. The method was developed and evaluated using a modeling framework where all components were known (Nelson et al. 2018), and evaluated against other tranpsiration estimation methods using the FLUXNET network (Nelson et al. 2020).*

5. For all the sites, what does the spatial extent "look" like? How much data from EC towers had good representation with the at station pixel? Also, how did you account for edge effect (L161)? Or if the field is too small, do you take the pixel that contains information on other fields?

*For the LST data from MODIS the cutout size is fixed to the 1 km2 pixel in which the coordinates of the EC fall. For the reflectance-based MODIS predictors, we selected all pixels with a high correlation with the tower pixel if the tower pixel does not have sufficient good quality observations. Thus, we do only implicitly account for edge effects, and definitely take information from other fields. This is a pragmatic approach to aligning EC footprints and pixels of satellite datasets. Future efforts should indeed quantify the (mis-)match in different ecosystem characteristics found in the area representing the EC observations and the co-located satellite observations, and ideally account for them in the up-scaling procedure. FLUXCOM-X is suited to ingest this kind of information once available.*

6. In Figure 3, your comparison point (A-D) showed similar color in terms of the magnitude of NEE, highlighting the improvements in spatial and temporal resolution in your product.

- Why was northern France's NEE, from Le Harve to Calais to Rouen region, so different from the three products?
- For NEE units, can you add it is umol of CO2?

*Thanks for pointing out the incomplete unit in the plot. We have adjusted this. Regarding the*

*difference in magnitude in northern France, the temporal resolution may play a role here with the subdaily resolution being able to reproduce long daylight hours at this time of the year and hence a longer time with actual photosynthesis. But several other factors may result in this difference as well, including pure chance at this selected time period. The NEE magnitudes in Spain also show interesting differences between the products. Explainable AI methods may be a way forward to better understanding the predictions in different places and trace down the origin of differences between products.*

7. In S3.2 and on, you used Pg C and km3, I don't doubt that your calculation is correct. But these are different from units you listed in Table 2. How did you convert these?

*Indeed this was not clarified in the text. We have added a short paragraph at the end of the Data and Methods section outlining the units used in each case and briefly describing the conversion methods.*

8. Based on Figure 5, X-BASED underestimates compared to RS and RS+METEO. But each has different resolution, this is where considering a reference truth dataset would be helpful. Otherwise, what is the highlight of this product? Also, it is interesting to see X-BASE, a higher spatial/temporal product, had less visible trends, any explanations?

*Thank you for these comments. In this comparison we aggregated all products to a consistent resolution, unit and land mask in order to achieve a fair and meaningful comparison. We refrained from defining a reference truth dataset because we think that no such data set exists and that rather they are complementary with individual strengths and weaknesses. Instead, we are interested in the qualitative and quantitative differences to understand which spatio-temporal patterns may be robust and reliable in the different datasets. X-BASE products are certainly far from perfect, but one highlight compared to FLUXCOM products is certainly the regional to global uptake magnitude which is more consistent with independent top-down estimates from inversions.*

*Regarding the smaller trend in X-BASE NEE than in FLUXCOM again we can only speculate. Possibly the trends in the predictors play an important role here. RS has a trend towards more negative NEE and can only be driven by trends in satellite observations, while RS+METEO (which only can show trends due to trends in meteorological variables) shows a trend towards less uptake, so in X-BASE possibly some of these effects compensate.*

9. One thing I am missing is that you compared X-BASE with other products but short on why there may be differences. For example, in L284, why was the RS GPP lower than the two other products?

*Thank you for this question, and we have asked ourselves the same question actually, but a definitive and detailed answer as to what is the main cause of the magnitude difference needs further investigation in the form of experiments. These are generally possible within Fluxcom-X, but have not been the scope of this work. Figure 6 shows that both RS and RS+METEO have lower GPP than X-BASE in temperate and boreal northern hemisphere ecosystems. Unlike RS, RS+METEO has considerably higher average GPP in the evergreen tropics which will balance the lower GPP in the middle and higher latitudes and results in similar global GPP like X-BASE. Conversely for RS, which may be affected by the absence of meteorological predictors in the tropics. Again, these are only speculations, but we look forward to analyzing the direction and magnitude of the effects on the flux estimates of temporal resolution and predictor sets. For clarification, we have added a general paragraph at the beginning of the discussion section:*

*We have presented qualitative and quantitative comparisons of the X-BASE products with previous FLUXCOM estimates and independent data sets where possible. Statements on the main causes of (dis-)agreements in the magnitude and spatio-temporal variability in the flux estimates remain speculative as detailed analyses would require further investigation in the form of experiments, which is beyond the scope of this manuscript. In the following, we discuss the prominent changes in NEE compared to FLUXCOM as a key scientific highlight in X-BASE and outline potential ways forward to tackle persistent challenges in empirical upscaling.*

10. You mentioned the performance differences in semi-arid regions a few times in S3.2.3, I would like to see more discussions around this, even with an example place.

*Yes we agree that patterns regarding water fluxes in water limited regions is an aspect that warrants much further exploration. In the case of water fluxes, we only compare to GLEAM as an independent reference as it is also model output thus not a ground truth. As such, we would like to avoid over analyzing our results or drawing conclusions beyond what the analysis shows. We are currently working on a more in-depth analysis using the FLUXCOM-X framework to test the water fluxes more in depth. However, we have added some further discussion on semi-arid regions in the discussion.*

11. In L375, what do you mean by enhanced EC quality? Is this achieved by ONEFLUX, or did you do additional procedures like a spectral correction? You also mentioned biases of EC in L385, and this is where review on EC uncertainty would be nice.

*While it is clear that the improvements come from the eddy covariance data, it is hard to pin-point a specific change as there were many improvements from the La Thuile dataset to the data used here. The eddy covariance processing pipeline is still at least partly ad hoc, so all spectral corrections and initial quality control is done by the tower teams before it even passes to ONEFLUX. A standardized spectral correction is currently not possible because the raw turbulence data are not consistently available in all flux releases. ONEFLUX was also implemented after the La Thuile dataset. Except cases like the ICOS and NEON networks where the fluxes calculation is centralized starting from the raw data and for this reason more standardized and documented. Furthermore, the general knowledge and experience with eddy covariance has increased over 15 years, so the improvements are really a community effort. We have clarified this paragraph to highlight this point, which complements the additional paragraph in the introduction on eddy covariance uncertainties (See response to #2).*

12. I like this part, but I have few comments:

   • could you list out the challenges and show the readers by point in terms of which is addressed in X-BASE and the next step.

*The challenges are partly interrelated and a way towards addressing them may involve a combination of factors rather than a single one. This will of course need to be verified by analyses to actually tackle the issues. Because of the inter-relatedness, we decided to outline the issues and solutions in text form, but have nevertheless rephrased the first part of this paragraph in an attempt to clarify this upfront. We have also rephrased individual sentences throughout to clarify on research priorities in FLUXCOM-X developments.*

   • For L418, what about in the field? Based on the results, what would you need? What about data availability? There are key regions that do not have towers yet whether because financial or practical constraints.

*Regarding EC data, any additional site or site-year is valuable, as has been demonstrated by the marked change in X-BASE global NEE to FLUXCOM. Our results furthermore suggest that subtropical agricultural sites are underrepresented, and that also dry ecosystems along temperature gradients would strongly benefit from better EC coverage. Detailed information on which conditions are underrepresented could be inferred from an approach of quantifying how the environmental conditions at any place are presented by the conditions sampled in the training data set (Papale et al. 2015, Jung et al. 2020, SI, Pallandt et al. 2022, both in Biogeosciences). Disturbance conditions and recovery from disturbance is severely underrepresented in the eddy covariance data everywhere.*

- L422, on the use of SAR and LiDAR, sure, but what about thermal platform like Landsat and ECOSTRESS and multispectral platform like Sentinel?

*Certainly also other platforms like the ones you mentioned have considerable potential to improve flux estimates, but each of them also has their specific challenges. Landsat is promising because of high spatial resolution and the potential of accurately matching EC fetches and satellite pixels/foorprints but disadvantageous in time. ECOSTRESS has geolocation issues which will become very important at high resolutions and does not cover the higher latitudes. The Sentinels are a promising continuation for MODIS and offer high spatial detail, but have comparatively low temporal overlap with the majority of available EC data. An optimal combined usage of several of these spaceborn platforms requires methodological developments, which is clearly one of the priorities in future FLUXCOM-X developments. For reasons of clarity we wrote this paragraph in a general way without mentioning individual satellite missions or data sets, but have added a sentence in the previous paragraph that mentions the need for methodological developments to optimally (and jointly) ingest different Earth observation data streams into empirical up-scaling.*

**Other comments:**

- Please add short description on the ONEFLUX data processing (L104)

*This has been added.*

- How much of your data were removed prior to gap filling (L111)?

*The final filtering and gap-filling steps of the eddy covariance data was performed by the ONEFLUX software before we have access to it. By estimating the number of values marked as original and not gap filled, we find that around 50-65% can be gap filled in the original dataset depending on the flux. In particular, NEE has additional filtering for low turbulance conditions (USTAR filtering) that can result in many night time values with stable conditions to be filtered. These numbers also include padded years, as the dataset always starts on January 1st and ends on December 31st, as well as instrument down time. Overall, the number of gap filled data is consistent with standard usage of FLUXNET data and previous synthesis work.*

- In L121, one of the QC is based on gap-filled confidence. Do the original data come with gap-filled variables? Or you are performing gap-filling here? If so, how?

*Yes the original flux data is gap-filled using the MDS algorithm in the ONEFLUX pipeline, which has been clarified in the text. In the case of meteorological data, gap filling is performed using a downscaling from ERA5, but only periods where all variables were measured were used in training.*

- For S2.2, did you cross validate your ERA5 variables? How did you use a 2m TA for a tower that may be taller than that?

*The ERA5 data was used for the global data, so is not expected to be representative of any one tower. We did however test that there was no overall bias when comparing the meteorological variables at site and global level.*

- Was cloud coverage a concern for quality checks (L145)?

*Yes! We have included this in the text.*

- Since you are using GEE, is MODIS v006 TERRA (L155) MOD11 or MOD21? If 11, did you see nighttime cold bias, and how did you address that?

*We used the MOD11 products, and did not check or account for a nighttime cold bias. The product, and especially also the product version (006) used, has deficiencies, including biases over arid regions due to the static emissivity assumption. Nighttime cold biases are also being discussed (e.g. https://tc.copernicus.org/articles/12/907/2018/). For the X-BASE version presented in the manuscript, however, MxD21 is not available for MODIS product version 006 in a climate modelling grid. Current work in progress towards updating to the most recent MODIS version 061 also includes the potential transition towards using LST from the MxD21 products. We have already acquired the MxD21 v061 data in order to test the effect that the different LST retrievals may have on the final product.*

- I am not too familiar with machine learning. For the training (L170), are these ratios low enough to prevent overfitting?

*Overfitting in the XGBoost algorithm used here is less sensitive to the subsampling rate, but rather the total number of boosting rounds. Here we tested the number of boosting rounds using a validation dataset of sites which are withheld during initial training. By leaving out entire sites, we check against the model overfitting to a single subset of sites and stop training once the performace as measuring on the validation set stops improving. Additional clarifications have been added to the text.*

- Check the units in Figure 5. Should the upper panel be Pg instead of g? For monthly mean, should it be just mass of C per unit area on y-axis and per unit time over x-axis? This second comment goes for the inter-annual variability unit as well.

*Here we show multiple timescales and tried to give units that are most relevant. The globally integrated numbers (lower panels) are in PgC, while the seasonal cycles are in average gC per day per square meter, as this unit is more often used in literature rather than gC per month per square meter, and is consistent with that reported in FLUXNET.*

- The sentence of "However..." from L259-262 reads weird, please rephrase.

*Indeed. Done.*

- Figure 7 is slightly hard to see. Please consider resizing as well as renaming the plot titles clearly, like X-BASE vs. TROPOMI.

*Done.*

- What do you mean by transport of water laterally in L333? Is this in the atmosphere? Or runoff? Or something else?

*Here we mean runoff, this has been clarified in the text.*

**Response to reviewer 2**

**Overview:**

This paper mainly presents X-BASE, an ML-based upscaling product of carbon and water fluxes. The paper is well-written and clear, presenting important and interesting information.

*Thank you for the kind words and taking the time to review this manuscript.*

**Minor concerns:**

- Line 76: using the word "quality" twice seems to be unnecessary.

*The word "quality" appears twice by referring to EC data once and then to the predictors on the other hand. We agree this sentence was not fully clear and have reordered the sentence to increase clarity.*

**Major concerns:**

- Table 2: why do you use per-second units for NEE and GPP? Why not gC per hour?

*Here the units are consistent with those published in FLUXNET, where sub-daily values are in micromole and daily or coarser units are in grams carbon. Of course different use cases will require different units. This has been clarified in a new section at the end of the Data and Methods section.*

- Section 2.4: although X-BASE is an ML-based product, the explanation of the ML model is too brief. For example, it is not clear how the hyperparameters of the XGBoost model were determined or optimized. To ensure the reproducibility of the methodology, this section should be modified.

*Indeed this section was not complete enough to fully recreate the model. Hyperparameters were chosen based on preliminary testing. As model development was not a main focus of this paper, we used XGBoost, which is relatively robust, with the idea to focus further works specifically on the machine learning elements. However, we have expanded the description of the model, including a full list of the hyperparameters used in the supplementing information.*

**Response to editor's comments**

I read through myself and agree it is a dataset update of interest to the community. I have a few additional comments:

1) I agree with Referee #1 that an explicit statement/paragraph about what is different from FLUXCOM is needed (like spatial resolution, a new ET product, etc.).W

*As noted in the response to Reviewer 1, we have rewritten parts of the introduction, discussion, as well as added context in supplementary materials to highlight these differences.*

2) In figure 1, can explicit numbers be provided for how much samples increased?

*We were a bit hesitant to provide these numbers, as there is not a direct comparison and we would not like overstate how the number of points in the dataset might be contributing to model performance, but rather focus on where we are sampling. However, we have added these number in the associated results section:*

> *Overall for NEE, the number of sampled site-days increased over three-fold (552878 to 183216 for X-BASE and RS+METEO respectively), note however that X-BASE is modeled at hourly instead of daily resolution and thus the number of sampled site-days should not be considered a metric of how well the feature space is sampled.*

3) A discussion of spatial scale mismatch of towers and gridded products is needed

*Yes, this is a key aspect to all up-scaling exercises, and one that we hope the FLUXCOM-X framework will be able to better diagnose. We have added further discussion on this topic the "Tackling persistent challenges".*

---

## Author Response (AR2)

**Response to reviewer and editor comments**

Thank you to all reviewers and editors who have taken the time to improve our manuscript. We have addressed all point suggested.